# Full-length direct RNA sequencing uncovers stress granule-dependent RNA decay upon cellular stress

Showkat Ahmad Dar[1†], Sulochan Malla[1†], Vlastimil Martinek[1,2,3], Matthew John Payea[1], Christopher Tai-Yi Lee[1], Jessica Martin[1], Aditya Jignesh Khandeshi[1], Jennifer L Martindale[1], Cedric Belair[1], Manolis Maragkakis[1]*

[1]Laboratory of Genetics and Genomics, National Institute on Aging, Intramural Research Program, National Institutes of Health, Baltimore, United States; [2]Central European Institute of Technology, Masaryk University, Brno, Czech Republic; [3]National Centre for Biomolecular Research, Faculty of Science, Masaryk University, Brno, Czech Republic

*For correspondence:
maragkakis@nih.gov

†These authors contributed equally to this work

Competing interest: The authors declare that no competing interests exist.

## eLife Assessment

This **important** study describes mRNA shortening during cellular stress and interestingly observes that this shortening is dependent on localization in stress granules. Surprisingly, this mRNA shortening does not appear to require the shortening of poly A tails. These are novel, paradigm-shifting findings, using cutting-edge technologies and **convincing** data, that should be of broad interest to the RNA community and beyond.

**Abstract** Cells react to stress by triggering response pathways, leading to extensive alterations in the transcriptome to restore cellular homeostasis. The role of RNA metabolism in shaping the cellular response to stress is vital, yet the global changes in RNA stability under these conditions remain unclear. In this work, we employ direct RNA sequencing with nanopores, enhanced by 5' end adapter ligation, to comprehensively interrogate the human transcriptome at single-molecule and -nucleotide resolution. By developing a statistical framework to identify robust RNA length variations in nanopore data, we find that cellular stress induces prevalent 5' end RNA decay that is coupled to translation and ribosome occupancy. Unlike typical RNA decay models in normal conditions, we show that stress-induced RNA decay is dependent on XRN1 but does not depend on deadenylation or decapping. We observed that RNAs undergoing decay are predominantly enriched in the stress granule transcriptome while inhibition of stress granule formation via genetic ablation of G3BP1 and G3BP2 rescues RNA length. Our findings reveal RNA decay as a key component of RNA metabolism upon cellular stress that is dependent on stress granule formation.

## Introduction

Cells respond to stress by triggering extensive transcriptome remodeling to restore cellular homeostasis (*Galluzzi et al., 2018*; *Payea et al., 2023*). Cellular stress responses have been under intense study as the capacity of cells to activate adaptive pathways deteriorates with age (*Derisbourg et al., 2021*; *Haigis and Yankner, 2010*; *Maragkakis et al., 2023*) and has been associated with many diseases, particularly age-related neurodegeneration (*Beckman and Ames, 1998*; *Stadtman and Berlett, 1997*).

The response to different stressors converges at the induction of the integrated stress response (ISR), a common adaptive molecular pathway (*Harding et al., 2003*; *Pakos-Zebrucka et al., 2016*) that induces phosphorylation of eukaryotic translation initiation factor 2 alpha (eIF2α) to inhibit translation initiation and decrease global protein synthesis (*Hinnebusch, 1994*). In turn, mRNAs exiting the translational pool get localized in stress granules (SGs), membraneless organelles that are formed upon stress when translation of mRNAs gets arrested (*Panas et al., 2016*; *Protter and Parker, 2016*). SGs comprise several different types of biomolecules, particularly mRNAs and RNA-binding proteins, but their precise functions in mRNA storage or decay are not yet fully understood (*Anderson and Kedersha, 2008*; *Marcelo et al., 2021*).

Short- and long-read sequencing following tagging of RNA 5′ ends has revealed that RNA content in cells is a mix of intact RNA molecules and RNA decay intermediates that exist in a transient, partially degraded state (*Ibrahim et al., 2021*; *Ibrahim et al., 2018*; *Pelechano et al., 2015*). However, much less is known about the global transcriptome-wide state of RNA under stress and whether stress leads to RNA decay or stabilization. Evidence has been found in both directions, with stress having been associated with mRNA stabilization in a translation-dependent and -independent way (*Gowrishankar et al., 2006*; *Hilgers et al., 2006*; *Horvathova et al., 2017*; *Kharel et al., 2023*). On the other hand, in yeast, ribosomal and other RNA-binding proteins, which under certain physiological conditions contact mRNAs to facilitate translation, show diminished mRNA association under heat shock stress and decreased mRNA abundance mediated by the 5′–3′ exoribonuclease Xrn1 (*Bresson et al., 2020*). Similarly, under oxidative stress, deletion of Xrn1 can lead to the accumulation of oxidized RNAs bearing nucleotide adducts such as 8-oxoguanosine (8-oxoG), in turn inhibiting translation and triggering No-Go decay (*Yan et al., 2019*).

Here, we globally profile the integrity of individual RNA molecules in human cells under stress by full-length direct RNA sequencing with nanopores. We develop a new computational framework for statistical interrogation of RNA decay at single-molecule and -nucleotide resolution. We find that upon cellular stress RNAs globally exhibit marked shortening at their 5′ end. This stress-induced decay is mediated by XRN1, but is not dependent on prior deadenylation or decapping, as classical models would posit (*Passmore and Coller, 2022*). Importantly, we find that the decaying RNA molecules are preferentially enriched in the SG transcriptome. Prevention of SG formation by ablation of cells for G3BP1 and G3BP2 rescues RNA length upon stress, suggesting SG formation as a required step for stress-induced RNA decay. Our results identify 5′ RNA decay as a hallmark of cellular stress response that defines the transcriptome and that is dependent on SG formation.

## Results

### Stress induces widespread decay at the 5′ end of RNAs

To explore the state of RNA under stress, we treated HeLa cells with sodium arsenite (0.5 mM) for 60 min to induce oxidative stress. We verified the formation of SGs and increased eIF2α phosphorylation (discussed in subsequent sections) and monitored cell viability up to 4 hr after treatment with no major change observed (*Figure 1—figure supplement 1a*). We collected arsenite-treated and untreated cells and performed direct RNA-seq with the TERA-seq protocol (*Ibrahim et al., 2021*; *Figure 1a*). Direct RNA-seq profiles RNA at single-molecule resolution without cDNA conversion or PCR amplification (*Workman et al., 2019*) while TERA-seq additionally incorporates unique adapters that are ligated to the 5′ phosphorylated end of individual RNA molecules. These adapters are then sequenced along with the native RNA sequence. Nanopore-based direct RNA sequencing proceeds in the 3′ to 5′ direction; thus. the inclusion, and subsequent computational identification of an adapter, at the 5′ end of a read acts as internal control, ensuring an RNA molecule has been sequenced in its entirety. Notably, TERA-seq does not biochemically select for ligated molecules; thus, the resultant sequencing datasets comprise both ligated and non-ligated molecules. Ligated molecules can be computationally recognized and separated to be analyzed jointly or independently to nonligated reads. Samples were prepared in biological triplicates and sequenced on a MinION device, yielding approximately 2 million long reads per library (*Supplementary file 1*). Replicates displayed high expression correlation and high alignment efficiency to the human genome (*Figure 1—figure supplement 1b*, *Supplementary file 1*). Consistent with oxidative stress induction, differential gene expression analysis identified several stress response genes being upregulated in arsenite while

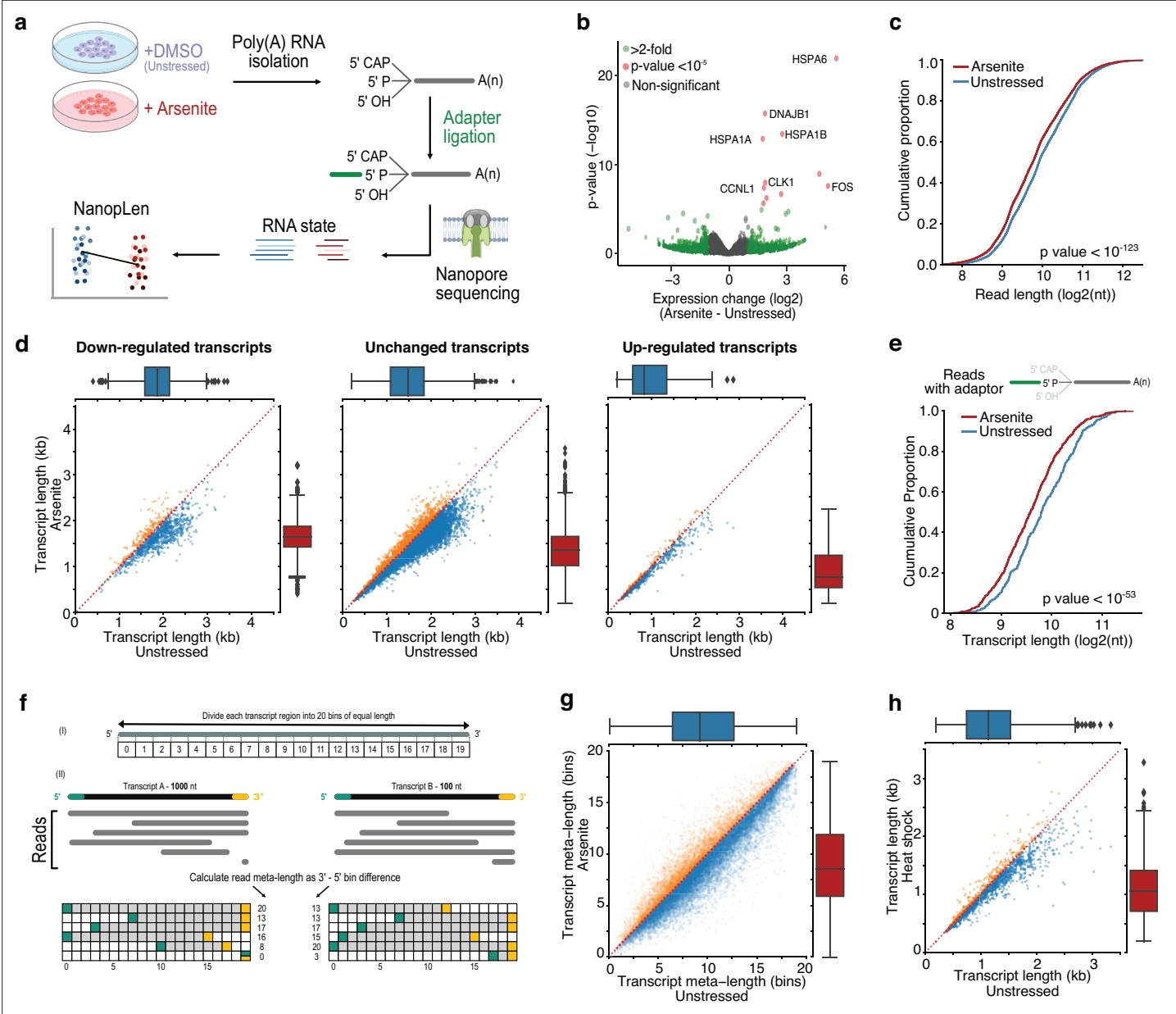

**Figure 1.** RNA shortening upon cellular stress. (**a**) Schematic of experimental design (**b**) Volcano plot for the differential expression of arsenite-treated and unstressed cells. Green color indicates genes with more than two-fold difference and red indicates statistical significance p-value<10⁻⁵. (**c**) Cumulative distribution of read length for arsenite-treated and unstressed cells. (**d**) Scatter plots of average transcript length for arsenite-treated and unstressed cells stratified by their differential expression change. Downregulated: (-Inf, –0.5), unchanged: (–0.5, 0.5), upregulated (0.5, Inf) fold-change. Only transcripts with at least five aligned reads are shown. Red dotted indicates the y = x line. Color indicates transcripts below (blue) and above (orange) the diagonal. (**e**) Cumulative distribution of transcript length for arsenite-treated and unstressed cells using only reads with adapter ligated at the 5′ end. (**f**) Schematic of read meta-length calculation. Each annotated transcript is divided into 20 equally sized bins. Each read is then assigned meta-coordinates depending on the bin in which its 5′ and 3′ templated ends align. The read meta-length is calculated as the difference of the meta-coordinates and presented as a percentage of full length. (**g**) Scatter plot of average transcript meta-length for arsenite-treated and unstressed cells. Coloring is the same as (**d**). (**h**) Scatter plot of average transcript length for heat shock and unstressed cells.

The online version of this article includes the following figure supplement(s) for figure 1:

**Figure supplement 1.** RNA shortening upon cellular stress.

gene ontology enrichment analysis of differentially expressed genes showed expected associations to cellular response to stress (*Figure 1b*, *Figure 1—figure supplement 1c*, *Supplementary file 3*). These results show that our long-read data reproducibly captured the expression status of the transcriptome upon oxidative stress.

Surprisingly, quantification of read lengths revealed a significant difference in read lengths with RNAs from oxidatively stressed cells being significantly shorter than unstressed cells by an average of 115.3 nucleotides (Mann–Whitney: p-value$<10^{-123}$, *Figure 1c*). To test this finding for individual transcripts ('transcript' hereafter refers to the annotated loci of a gene isoform), we calculated a metric for transcript length corresponding to the average length of reads of each transcript. We again found that RNAs from stressed cells were globally shorter than in unstressed cells, independent of both transcript differential expression (*Figure 1d*) and coding potential (*Figure 1—figure supplement 1d and e*). To exclude potential library preparation artifacts or failure to capture long transcripts by the sequencing device, we repeated the analysis by computationally selecting only reads with ligated 5′ adapters, ensuring RNA molecules were fully sequenced. The subset of 5′ adapter-ligated reads confirmed our findings and again showed global RNA shortening upon oxidative stress (*Figure 1e*). Since both ligated and nonligated reads showed comparable results, all reads were used in subsequent analysis to increase sequencing depth, unless otherwise mentioned.

To test whether the observed stress-induced RNA shortening was independent of pre-existing steady-state fragmentation levels, we performed analyses in a length-normalized space (meta-length) that represents read length as a percent of annotated transcript length. We calculated normalized meta-length by splitting each annotated transcript in 20 equal bins, assigning the mapped read ends into individual bins, and defining the transcript meta-length as the difference of the 5′ and 3′ end bins (*Figure 1f*). Our data showed a consistent reduction of transcript meta-length upon oxidative stress, independent of steady-state unstressed fragmentation levels (*Figure 1g*, *Figure 1—figure supplement 1f*). This finding remained consistent when only selecting reads with an identified poly(A) tail (*Figure 1—figure supplement 1g*).

To exclude potential confounding effects of arsenite itself, we further tested oxidative stress induction with 0.3 mM hydrogen peroxide ($H_2O_2$) for 2 hr followed by TERA-seq. $H_2O_2$-induced oxidative stress also resulted in significant transcript shortening (Mann–Whitney: p-value$<10^{-96}$), similar to that observed with arsenite, indicating that oxidative stress results in RNA shortening irrespective of inducer (*Figure 1—figure supplement 1h*). To explore whether RNA shortening is unique to oxidative stress, we re-examined publicly available direct RNA-seq data for human K562 cells that were subjected to heat shock at 42°C for 60 min (*Maier et al., 2020*). Our results showed that heat-shocked cells also had significantly shorter RNAs compared to unstressed cells (Mann–Whitney: p-value$<10^{-76}$), again independently of differential expression (*Figure 1h*, *Figure 1—figure supplement 1i and j*). Additionally, we also observed RNA shortening in mouse embryonic fibroblasts 3T3 cells treated with sodium arsenite, suggesting an evolutionary conserved process (*Figure 1—figure supplement 1k and l*).

To identify transcripts with statistically significant RNA length changes, we developed NanopLen, a tool that uses linear mixed models, and modeled the library as a random effect to adjust for variation across replicates. To test the model, we simulated sequencing data over varying RNA shortening proportions and expression counts. The simulated data showed that the model accurately predicted the true length difference for each tested shortening proportion and had a well-controlled false-positive rate (*Figure 2a*). As expected, NanopLen reported lower significance for smaller length differences while higher expression counts resulted in higher statistical power on simulated data. We subsequently employed NanopLen to compare unstressed and arsenite-treated cells and identified 2730 significantly shortened transcripts (p-value$<0.05$) (*Figure 2b*, *Supplementary file 4*). Gene ontology analysis of these most highly shortened transcripts showed processes relevant to RNA catabolism, translation initiation, and protein localization to the endoplasmic reticulum (*Figure 2c*).

The presence of a poly(A) tail is required for direct RNA-seq. Therefore, we anticipated that our method would only be able to capture fragmentation occurring at the 5′ end of RNAs. As expected, significantly shortened transcripts showed that their 3′ templated end was almost identical in oxidative stress and unstressed conditions, whereas the corresponding 5′ end was heterogeneous (*Figure 2d*, *Figure 2—figure supplement 1a*). Quantification at the transcriptome-wide level further confirmed this observation (Mann–Whitney p-value$<10^{-256}$, *Figure 2—figure supplement 1b*). To validate our

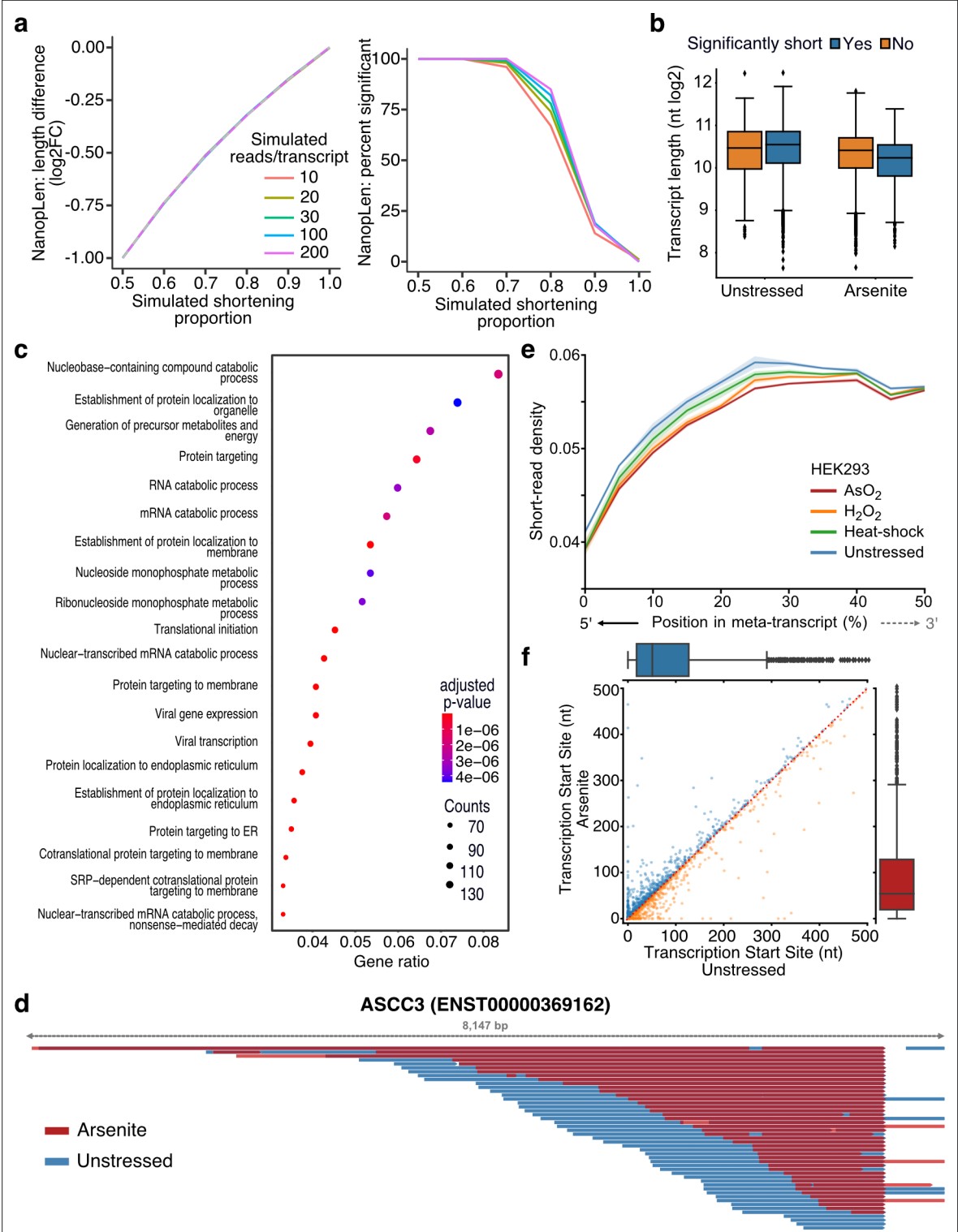

**Figure 2.** Characterization of stress-induced shortened RNAs identified by NanopLen. (**a**) Left: line plot of average length difference estimate for simulated data of varying read depths. True value is depicted by dashed grey line. Right: percentage of simulated genes that are detected as significantly different in length. The null simulation of no shortening corresponds to simulated shortening proportion 1.0. (**b**) Box plot of average transcript length for statistically significant and nonsignificantly shortened transcripts identified through differential length analysis in arsenite-treated and unstressed cells using NanopLen. (**c**) Gene ontology analysis for biological processes of significantly shortened transcripts. (**d**) IGV screenshot of ASCC3 aligned reads for arsenite-treated (red) and unstressed cells (blue). Libraries were randomly downsampled to maximum 50 reads per window

*Figure 2 continued on next page*

Figure 2 continued

and the libraries were overlayed. All reads were used, irrespective of adapter ligation status. (**e**) Short-read RNA-seq density at the 5′ half of transcripts for unstressed, NaAsO$_2$, H$_2$O$_2$ and heat shock-treated HEK293 cells. Shade indicates standard error of the mean for replicates. (**f**) Scatter plot of transcription start site position for arsenite-treated and unstressed HeLa cells.

The online version of this article includes the following figure supplement(s) for figure 2:

**Figure supplement 1.** Characterization of stress-induced shortened RNAs identified by NanopLen.

sequencing results, we performed RT-qPCR analysis using pairs of primers targeting regions along each transcript. We found that arsenite-treated cells had substantially reduced amplification from primer pairs closer to the 5′ end of the transcript than near the 3′ end compared to control cells (*Figure 2—figure supplement 1c*). We reasoned that RNA shortening at the 5′ end should also be reflected in short-read RNA-seq as reduction in read coverage at the 5′ end of RNAs. We thus reanalyzed data from HEK293 cells treated with a battery of stressors, that is, 42°C heat shock for 1 hr; 0.6 mM H$_2$O$_2$ for 2 hr; and 300 µM NaAsO$_2$ for 2 hr followed by short-read RNA-seq (*Watkins et al., 2022*). Our analysis showed that, compared to unstressed, all stress conditions resulted in significantly reduced read coverage at the 5′ end (*Figure 2e*, *Figure 2—figure supplement 1d*).

Finally, we considered the possibility that selection of alternative transcription start sites (TSSs) downstream of annotated TSSs could be contributing to RNA shortening. We have previously established that positions with high read density upstream of coding sequences in TERA-seq accurately represent TSSs matching those defined by CAGE-seq (*Ibrahim et al., 2021*). We used this method to independently calculate TSSs in stress and unstressed conditions in our HeLa data. Our results showed nonsignificant changes between conditions (Mann–Whitney: p-value=0.35) with no preference toward shorter or longer transcripts (*Figure 2f*), indicating that the observed shortening is not affected by TSS selection. Collectively, our results, although not excluding 3′ end decay for molecules not sequenced due to lack of a poly(A) tail, provide strong evidence for stress-induced decay occurring at the 5′ end of RNA molecules harboring poly(A) tails.

## Stress-induced RNA decay is XRN1-mediated but independent of deadenylation

XRN1 is the primary 5′ to 3′ exonuclease in cells, and thus a likely candidate for mediating 5′ transcript shortening upon stress following decapping or endonucleolytic fragmentation. We hypothesized that in the absence of XRN1, RNAs would be stabilized, and their observed length would be restored. Silencing of XRN1 with siRNAs (si*XRN1*) resulted in substantial reduction of protein level compared to non-targeted control (siCTRL) (*Figure 3a*, *Figure 3—figure supplement 1a*). We subsequently performed TERA-seq for siCTRL- and si*XRN1*-transfected cells in the presence or absence of oxidative stress. Our results showed that while arsenite and unstressed siCTRL cells had the highest transcript length difference (Mann–Whitney, p-value<10$^{-307}$), silencing of XRN1 largely abolished this difference (Mann–Whitney, p-value=0.18) and brought transcript lengths in line with unstressed cells with XRN1 silenced (*Figure 3b*). Similar results were observed for adapter-ligated transcripts, with XRN1 silencing showing an even stronger effect, suggesting that stress-induced decay depends on XRN1 (*Figure 3c*). While silencing of *XRN1* might be expected to have a general effect on RNA length, our results showed that RNA length rescue was predominant and specific for the previously identified stress-induced significantly shortened RNAs identified by our NanopLen analysis (*Figure 3d*). These results strongly implicate XRN1 to transcript shortening during stress.

Under traditional decay models, deadenylation is considered the first and rate-limiting step prior to decapping and subsequent exonucleolytic action from the 5′-end via XRN1 (*Borbolis and Syntichaki, 2022*; *Łabno et al., 2016*; *Mugridge et al., 2018*; *Passmore and Coller, 2022*). However, the RNA molecules we sequenced and found shortened still had a poly(A) tail, as this is required for dRNA-seq. This argued that removal of the poly(A) tail was not necessary for stress-induced 5′ end shortening. To systematically assess the role of deadenylation and decapping, we first downregulated the CCR4-NOT deadenylation complex by expressing a GFP-fused catalytically inactive form of the CCR4-NOT transcription complex subunit 8 (NOT8* D40A, E42A) (*Chang et al., 2019*) in HeLa cells, followed by arsenite treatment and dRNA-seq. Expression of the dominant negative NOT8* form was confirmed by epifluorescence (*Figure 3—figure supplement 1b*) and resulted in an expected increase in poly(A)

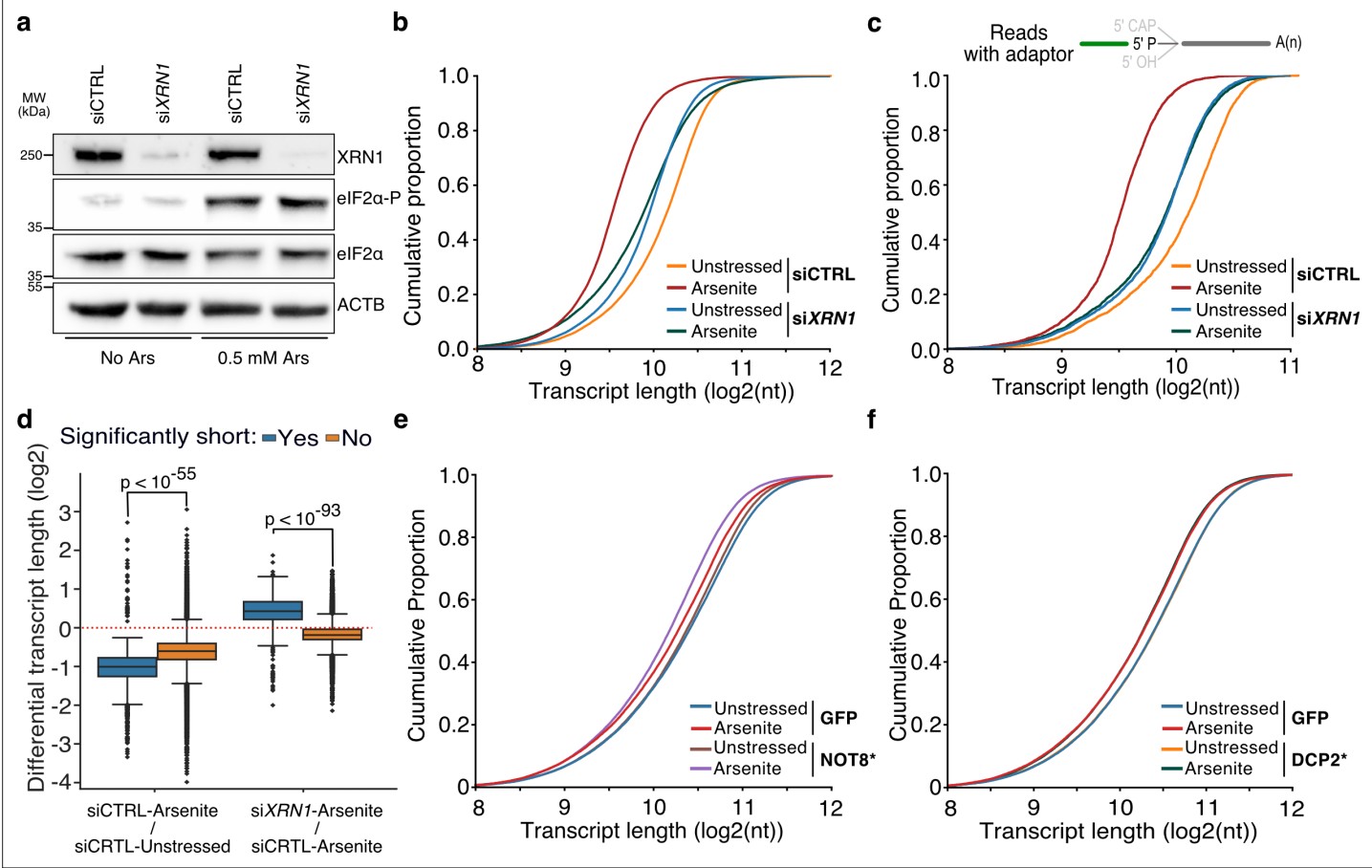

**Figure 3.** The effect of XRN1 knockdown on RNA shortening. (**a**) Immunoblot for XRN1, eIF2α-P, and eIF2α for cells transfected with a non-targeting (siCTRL) or *XRN1*-targeting (si*XRN1*) siRNA. ACTB is used as control. (**b**) Cumulative distribution plot of transcript length for XRN1 knockdown (si*XRN1*) and control (siCTRL) cells with and without arsenite treatment (unstressed). (**c**) Same as (**b**) but only reads with ligated 5′ end adapter are used. (**d**) Box plots of differential transcript length in arsenite-treated versus unstressed cells for significantly and nonsignificantly shortened transcripts upon XRN1 knockdown and control. (**e, f**) Cumulative distribution plot of transcript length for NOT8* D40A E42A and GFP-expressing cells (**e**) or DCP2* E148Q and GFP-expressing cells (**f**) with or without (unstressed) arsenite treatment.

The online version of this article includes the following source data and figure supplement(s) for figure 3:

**Source data 1.** Original files for western blot analysis displayed in *Figure 3a*.

**Source data 2.** File containing original western blots for *Figure 3a*, indicating the relevant bands and treatments.

**Figure supplement 1.** The effect of XRN1 knockdown on RNA shortening.

**Figure supplement 1—source data 1.** Original files for western blot analysis displayed in *Figure 3—figure supplement 1*.

**Figure supplement 1—source data 2.** File containing original western blots for *Figure 3—figure supplement 1*, indicating the relevant bands and treatments.

tail length compared to control GFP cells (*Figure 3—figure supplement 1c*). As hypothesized, our results identified significant 5′ shortening upon stress following expression of the dominant negative NOT8* (Mann–Whitney, p-value<$10^{-299}$), suggesting that deadenylation by the CCR4-NOT complex was indeed not required for stress-induced 5′ RNA decay (*Figure 3e*).

These results raised the possibility that decapping may also not be required for stress-induced decay. To test this, we expressed a GFP-fused catalytically inactive form of the mRNA-decapping enzyme 2 (DCP2), DCP2* E148Q (*Loh et al., 2013*) in HeLa cells followed by dRNA-seq. Expression of DCP2* was confirmed by epifluorescence, though it was less pronounced than that of NOT8* (*Figure 3—figure supplement 1b*). Interestingly, transcripts from DCP2*-expressing cells presented a similar significant (Mann–Whitney, p-value<$10^{-127}$) 5′ shortening upon oxidative stress as GFP control cells, suggesting that the canonical decapping pathway is also not necessary for stress-induced RNA

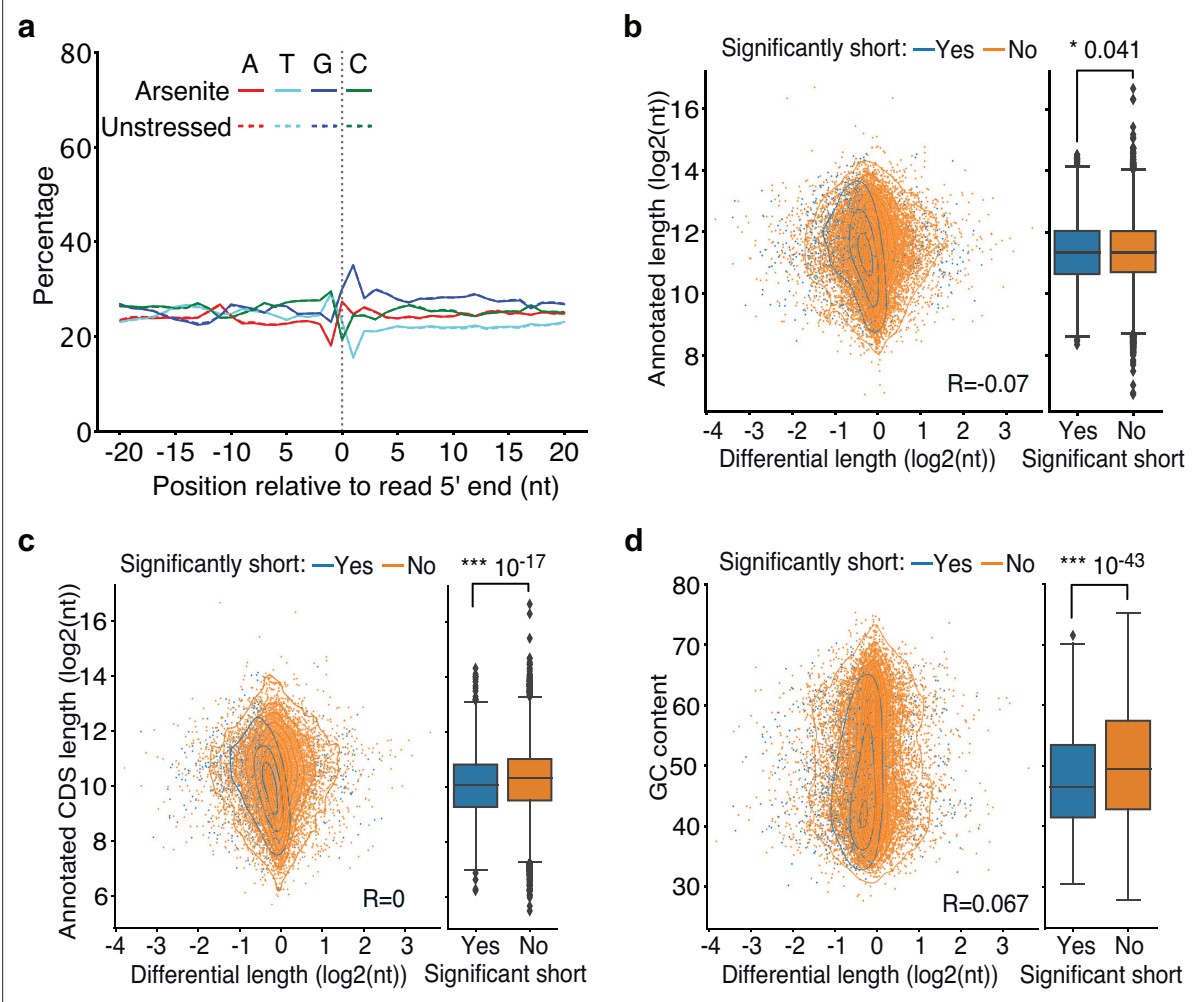

**Figure 4.** Association of poly(A) tail length and *cis*-regulatory elements with RNA shortening. (**a**) Nucleotide composition around the 5' end of reads in arsenite-treated and unstressed cells. All reads were used, irrespective of adapter ligation status. (**b–d**) Scatter plot of annotated transcript length (**b**), annotated coding sequence (CDS) length (**c**) and GC content (**d**) against transcript differential length in arsenite-treated and unstressed cells. The box plots on the right side summarize the y-axis variable for significant and nonsignificantly shortened transcripts. The Pearson's correlation coefficient and the Mann–Whitney *U* test p-value are shown.

The online version of this article includes the following figure supplement(s) for figure 4:

**Figure supplement 1.** Association of *cis*-regulatory elements and ribosome occupancy with RNA shortening.

decay, and thus likely involves an endonucleolytic cleavage event (*Figure 3f*). In conclusion, our results show that stress-induced RNA decay is mediated by XRN1 but is independent of prior deadenylation or decapping.

## Restoring ribosome density inhibits stress-induced RNA decay

The yeast Xrn1 has previously been associated with the decay of oxidized RNAs via the No-Go decay pathway following ribosome stalling at nucleotide adducts, particularly 8-OxoG, and endonucleolytic cleavage (*Yan et al., 2019*). We reasoned that if endonucleolytic cleavage induced by 8-OxoG was the major contributor to transcript shortening during oxidative stress, then an increase in guanine prevalence should be expected at the vicinity of 5' ends of sequenced RNAs upon stress. However, our data did not support this explanation as no G-nucleotide enrichment difference between arsenite and unstressed conditions was observed (*Figure 4a*). Similarly, no difference was observed upon XRN1 silencing (*Figure 4—figure supplement 1a*). These findings, combined with the presence of shortening under heat shock (*Figure 1h*), indicate that the stress-induced RNA decay described here is not mediated via 8-OxoG ribosome stalling.

To identify gene features that could be driving stress-induced RNA decay, we tested whether the annotated nucleotide content and transcript length were predictive of transcripts prone to stress-induced 5′ shortening. Our results showed a significant association with the annotated length, coding sequence length, and GC content, both being significantly lower for significantly shortened transcripts (*Figure 4b–d*) and only a minor association with the 5′ and 3′ UTR length (*Figure 4—figure supplement 1b and c*). Since the coding sequence is the primary region of ribosome occupancy, we tested whether pre-stress ribosome density could be defining 5′ shortening using publicly available ribosome profiling data from unstressed cells (*Park et al., 2016*). Our analysis found no association between ribosome density per RNA and stress-induced shortening (*Figure 4—figure supplement 1d and e*), indicating that the pre-stress ribosome levels on RNAs are not linked to RNA decay under stress.

Rather, another possibility could be that stress-induced RNA decay is associated with the rate at which RNAs leave the translation pool following inhibition of translation initiation upon stress. To test the role of translation initiation inhibition and ribosome run-off upon stress on stress-induced decay, we treated cells with the ISR inhibitor (ISRIB) that bypasses the effect of eIF2α phosphorylation and facilitates translation initiation under stress (*Rabouw et al., 2019*; *Sidrauski et al., 2015*). We confirmed that ISRIB had no noticeable effect on the dose-dependent phosphorylation of eIF2α but inhibited the SG formation under stress, as expected (*Figure 5a–c*). Polysome fractionation following ISRIB addition showed a clear reduction of the monosome fraction, indicating partial recovery of translation initiation, as previously described (*Sidrauski et al., 2013*; *Figure 5d*). We then performed TERA-Seq on ISRIB-treated and control cells in the presence or absence of arsenite. Our results showed that compared to cells treated with arsenite only, ISRIB treatment resulted in a significant shift of RNA length toward longer molecules (Mann–Whitney, p-value<$10^{-156}$) essentially restoring transcript length to the level observed in unstressed cells (*Figure 5e*). Importantly, previously identified significantly shortened transcripts showed the greatest recovery of their length compared to nonsignificant ones (*Figure 5f*). To test whether this could be simply attributed to a possible length imbalance between up- and downregulated genes, we again plotted the average transcript length stratified by the differential expression status. Our data show that the observed shift in length is independent of differential expression status (*Figure 5—figure supplement 1a and b*). These results indicated that stress-induced RNA decay is associated with RNAs, leaving the translation pool following translation initiation inhibition upon stress.

As an alternative to ISRIB which modulates translation initiation, we also treated cells with cycloheximide to interrogate the dynamics of ribosome elongation and to prevent ribosome run-off. Cycloheximide blocks elongation and thus traps ribosomes on RNAs along with translation factors in polysomes, thus decreasing SG formation (*Jayabalan et al., 2021*; *Kedersha and Anderson, 2007*; *Takahashi et al., 2013*). We confirmed that treatment with cycloheximide also reduced the monosome fraction, although to a lower degree than ISRIB (*Figure 5—figure supplement 1c*). Interestingly, cycloheximide treatment also inhibited RNA decay, particularly for the most significantly shortened RNAs largely rescuing their length (*Figure 5g–i*). We again did not observe any association between the shift in length and differential expression (*Figure 5—figure supplement 1d and e*). Combined, our results indicate translation initiation inhibition and the exit of mRNAs from the translational pool as a critical step towards stress-induced decay.

## Inhibition of SG formation in cells devoid of G3BP1/2 rescues RNA decay

During stress, ribosome run-off and the exit of RNAs from the translation pool is accompanied by the formation of SGs (*Protter and Parker, 2016*). To test whether transcripts subject to stress-induced decay associate with SGs, we used publicly available data representing genes enriched in the SG transcriptome (*Khong et al., 2017*). As previously described (*Khong et al., 2017*), we found that SG-enriched genes were generally significantly longer, particularly in their coding and 3′ UTR sequences, (*Figure 6a*, *Figure 6—figure supplement 1a*) and less expressed than SG-depleted ones (*Figure 6—figure supplement 1b*). However, differential gene length analysis showed a significant difference in gene shortening dependent on SG enrichment status. Specifically, SG-enriched genes were found to be shortened at a significantly higher level upon oxidative stress than SG-depleted or nonlocalized RNAs for both HeLa and U-2 OS cells (*Figure 6b and c*, *Figure 6—figure supplement 1c and d*). Consistent with stress-induced gene shortening being associated with SG enrichment, SG-enriched

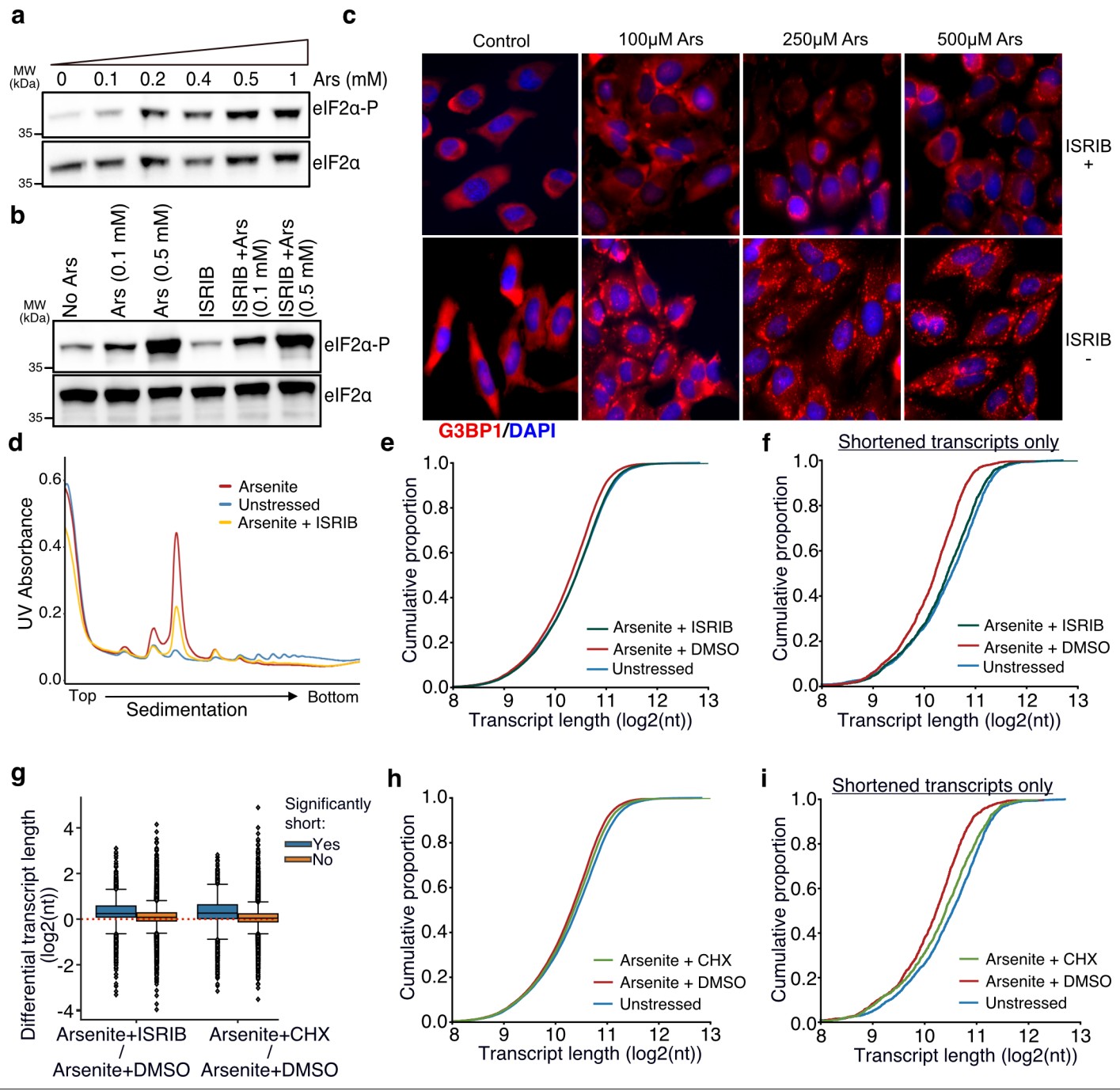

**Figure 5.** Translation and RNA shortening. (**a, b**) Immunoblot for eIF2α and eIF2α-P upon increasing concentration of arsenite (**a**) and cell treatment with 200 nM of ISRIB at different arsenite concentrations (**b**). (**c**) Stress granules (SGs) visualized by immunofluorescence of HeLa cells treated with indicated concentration of arsenite in the presence or absence of 200 nM ISRIB. A secondary goat anti-rabbit IgG H&L (Alexa Fluor 594) against G3BP1 (SG marker) and DAPI were used for visualization. (**d**) Ribosome sedimentation curve following cell treatment with ISRIB (200 nM) for arsenite-treated and unstressed cells. (**e**) Cumulative density plot of transcript length for arsenite-treated cells in the presence or absence of ISRIB. (**f**) Same as (**e**) for significantly shortened transcripts only. (**g**) Box plots of differential transcript length for comparisons indicated on the x-axis. (**h–i**) Same as (**e**) and (**f**) for cycloheximide CHX instead of ISRIB.

The online version of this article includes the following source data and figure supplement(s) for figure 5:

**Source data 1.** Original files for western blot analysis displayed in *Figure 5a and b*.

**Source data 2.** File containing original western blots for *Figure 5a and b*, indicating the relevant bands and treatments.

**Figure supplement 1.** Translation and RNA shortening.

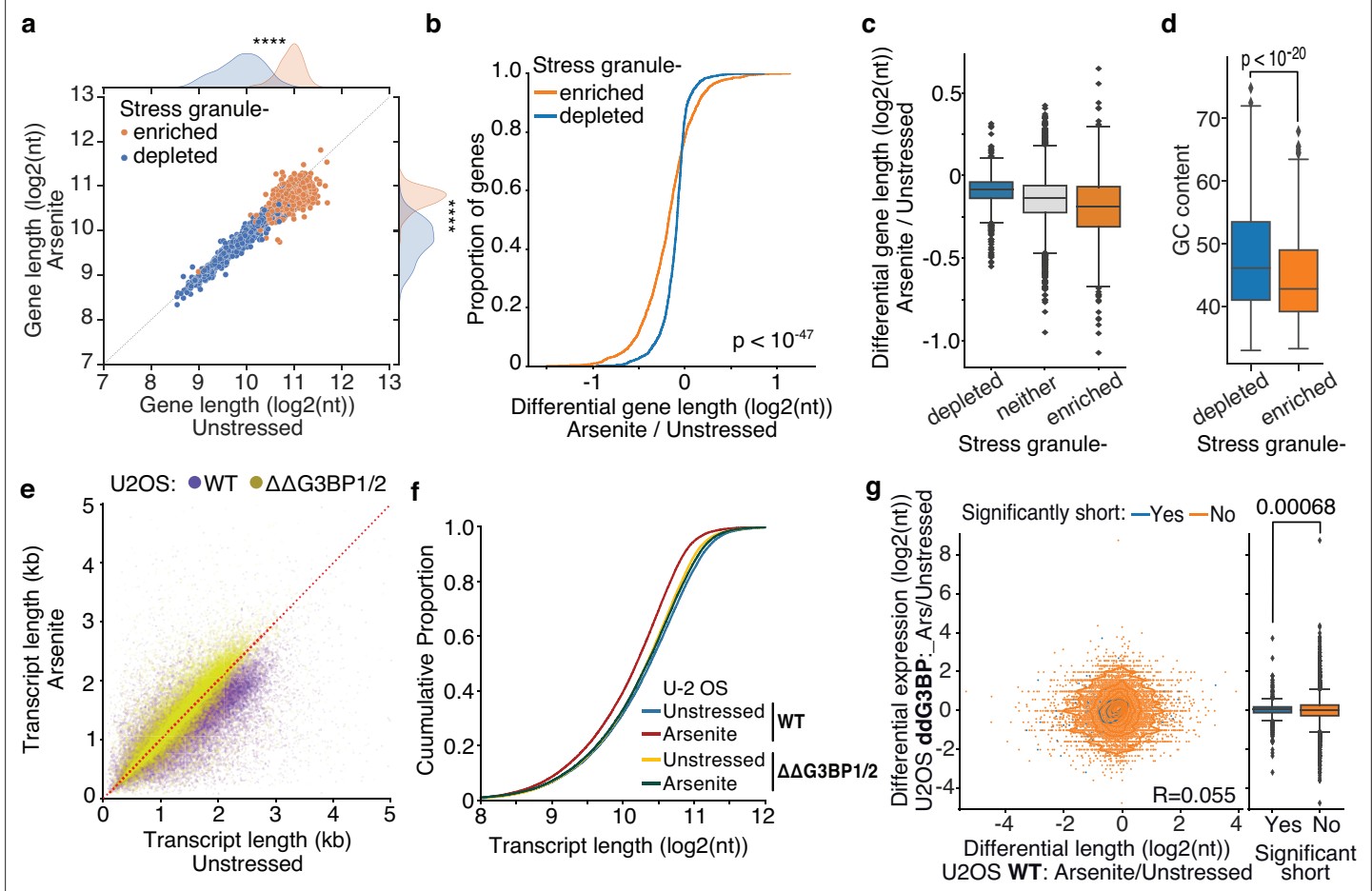

**Figure 6.** Inhibition of stress granule (SG) formation in cells devoid of G3BP1/2 rescues RNA decay. (**a**) Scatter plot of average gene length for arsenite-treated and unstressed cells stratified by gene SG localization. ****p-value<10⁻¹⁷². (**b, c**) Cumulative distribution and box plots of average gene length difference in arsenite-treated and unstressed cells stratified by gene SG localization. (**d**) Box plot of GC content percentage for SG-enriched and -depleted gene transcripts. (**e**) Scatter plot of average transcript length for arsenite-treated and unstressed U-2 OS and ΔΔG3BP1/2 U-2 OS cells. Only transcripts with at least five aligned reads are used. Red dotted indicates the y = x line. (**f**) Cumulative distribution plot of average transcript length in arsenite-treated and unstressed U-2 OS and ΔΔG3BP1/2 U-2 OS cells. (**g**) Scatter plot and box plot of differential transcript expression and shortening in arsenite-treated and unstressed ΔΔG3BP1/2 and WT U-2 OS cells.

The online version of this article includes the following figure supplement(s) for figure 6:

**Figure supplement 1.** Inhibition of stress granule (SG) formation in cells devoid of G3BP1/2 rescues RNA decay.

RNAs harbored lower GC content (*Figure 6d*), similar to significantly shortened transcripts (*Figure 4d*). To test whether the observed enrichment of shortened genes is specific to the SGs, we also tested for association with processing bodies (P-bodies), membrane-less organelles enriched in de-capping factors and exoribonucleases that are essential for RNA decay under normal, nonstress conditions (*Brothers et al., 2023*). We reanalyzed previously published data from HEK293 cells (*Matheny et al., 2019*), but in contrast to SGs, we found no association between P-body localization and gene lengths (*Figure 6—figure supplement 1e*). We also did not find an association between P-body localization and RNA shortening, which however could be a result of comparing different cell lines (*Figure 6— figure supplement 1f*). Our data show that shortened RNAs exhibit similar properties to SG-enriched transcripts and are preferentially enriched in the SG transcriptome.

To test the role of SGs in stress-induced RNA decay, we employed control (WT) U-2 OS and U-2 OS cells genetically ablated for both G3BP1 and G3BP2 (ΔΔG3BP1/2) via CRISPR/Cas9 (gift from Paul Anderson lab) previously developed and characterized in *Kedersha et al., 2016*. ΔΔG3BP1/2 cells inhibit SG condensation in response to arsenite and other stressors. Treatment of WT U-2 OS cells with arsenite showed a highly significant and global shortening of RNA upon stress (Mann–Whitney,

p-value<$10^{-307}$) in stark contrast to ΔΔG3BP1/2 cells that showed substantially less shortening (Mann–Whitney, p-value<$10^{-5}$) (*Figure 6e*, *Figure 6—figure supplement 1g and h*). Further comparison of ΔΔG3BP1/2 and WT U-2 OS showed that inhibition of G3BP1/2-mediated SG formation rescued RNA length to the level of unstressed WT cells (*Figure 6f*, *Figure 6—figure supplement 1i and j*). These results indicate that G3BP1/2-mediated SG formation is required for stress-induced RNA decay. Alternatively, SGs could be protective of shortened RNAs that otherwise would be rapidly eliminated from cells. Under this assumption, stress-shortened RNAs identified in WT cells would be expected to be depleted in ΔΔG3BP1/2 cells during stress. A comparison of differentially expressed transcripts upon arsenite treatment in ΔΔG3BP1/2 cells showed no association with WT shortening, indicating that in the absence of SGs these RNAs are not rapidly eliminated (*Figure 6g*), consistent with previous findings (*Bley et al., 2015*). Collectively our results show that G3BP1/2-dependent SG formation is required for stress-induced RNA decay.

## Discussion

Cellular response to stress is critical for cell recovery or induction of apoptosis if stress cannot be resolved. Recent works have delineated many of the biochemical pathways involved in stress responses. However, the full-length state of RNA under cellular stress remains incompletely characterized. In this work, we used TERA-seq, a protocol that involves the ligation of unique adapters at the 5′ end of RNAs (*Ibrahim et al., 2021*) to address a key limitation in nanopore direct RNA sequencing – the inability to consistently capture the 5′ end of sequenced RNA molecules. By sequencing RNA molecules end-to-end, we show that stress triggers decay at the 5′ end of RNAs. Interestingly our results show that stress-induced 5′ end decay leaves 'scars' that can also be identified in traditional short-read RNA-seq as decreased read density at the 5′ end of transcripts.

Our findings highlight XRN1 as an essential component for stress-induced decay, in the absence of which the RNA length is largely rescued. However, contrary to the traditional decay model, evaluation of the shortened RNAs did not reveal a dependency on deadenylation or decapping. Instead, our data point towards endonucleolytic cleavage as the most likely mechanism for the initial cut on RNA, generating either a 5′P directly recognized by XRN1 or a 5′OH that is subsequently phosphorylated to become a substrate for XRN1 (*Navickas et al., 2020*). Our analysis showing that transcript shortening is suppressed by treatments that increase ribosome occupancy (ISRIB and cycloheximide) also suggests that this putative endonucleolytic cut is likely not due to No-Go decay or a related ribosome quality control pathway since these decay mechanisms are guided to cleavage sites by ribosome collisions (*Doma and Parker, 2006*). However, it is currently unknown whether a mechanistic link exists with other co-translational decay pathways, such as ribothrypsis (*Ibrahim et al., 2018*), that also initiate by endonucleolysis. More studies will be needed to dissect the exact molecular events involved in this mechanism. Recent studies have found sporadic fragments of 3' UTRs as products of No-Go decay occurring at the sites of translation termination in oxidative stress, development, and brain aging (*Ji et al., 2021*; *Sudmant et al., 2018*). While most stress-induced decay fragments are longer than 3' UTRs and are not confined to translation termination sites, it is possible that some shorter fragments could also be contributing to an accumulation of 3' UTRs under these conditions.

Short coding sequences and low GC content were found as features of transcripts that are subject to stress-induced decay, arguing that there may be a component of structural stability to the specificity of 5' end shortening. Low GC content appears to also be a determinant for RNA recruitment to SGs, as also confirmed in our study (*Khong et al., 2017*). Consistent with these observations, our data show that the SG transcriptome is significantly enriched for stress-induced shortened RNAs. In fact, our results show that the formation of the SGs is required and indispensable for stress-induced decay as genetic ablation of G3BP1/2 rescues RNA length to the level of nonstressed controls. Inhibition of the ISR to reinitiate translation and dissolve SGs also curtails stress-induced RNA decay, further supporting a critical role for SG formation in stress-induced RNA decay perhaps through tethering with P-bodies, dynamically linked with SGs under stress (*Kedersha et al., 2005*; *Moon et al., 2019*).

The primary function of the ISR is to inhibit translation initiation and preserve energy for cell recovery during stress (*Pakos-Zebrucka et al., 2016*). While the physiological role of stress-induced RNA decay is currently unknown, it is intriguing to hypothesize that it evolved as a mechanism to further reduce translational load during stress. Endonucleolytic cleavage and removal of RNA 5′ ends could provide an orthogonal strategy to dial down translation, with several benefits: rapid elimination of translation

initiation sites, degradation of nonessential RNAs, release of ribosomes for stress response translation, and energy conservation. Future studies to test these hypotheses will be required, especially considering the emerging significance of SGs and RNA metabolism in neurodegenerative diseases and aging (*Ferrucci et al., 2022*; *Ripin and Parker, 2022*).

# Materials and methods

## Key resources table

| Reagent type (species) or resource | Designation | Source or reference | Identifiers | Additional information |
|---|---|---|---|---|
| Antibody | Beta actin mouse monoclonal | ProteinTech | Cat # 66009-1-Ig; RRID:AB_2687938 | (1:1000) |
| Antibody | Rabbit-anti XRN1 polyclonal | Thermo Fisher | A300-443A; RRID:AB_2219047 | (1:1000) |
| Antibody | Rabbit-anti eIF2α-P | Cell Signaling | Cat # 9721S; RRID:AB_330951 | (1:1000) |
| Antibody | Rabbit-anti eIF2α | Cell Signaling | Cat # 9722S; RRID:AB_2394335 | (1:1000) |
| Antibody | Goat anti-Rabbit IgG (H+L) Cross-adsorbed Secondary antibody | Thermo Fisher | Cat # 31462; RRID:AB_228338 | (1:10,000) |
| Antibody | Goat anti-Mouse IgG Fc Cross-Adsorbed Secondary Antibody, HRP | Thermo Fisher | Cat # 31439; RRID:AB_228292 | (1:10000) |
| Antibody | Anti-G3BP1 polyclonal | Thermo Fisher | Cat # PA5-29455; RRID:AB_2546931 | (1:500) |
| Antibody | Goat anti-rabbit IgG H & L (Alexa Fluor 594) | Abcam | Cat # ab150080; RRID:AB_2650602 | (1:200) |
| Cell line (*Homo sapiens*) | Human osteosarcoma, U-2 OS (control, WT) | Gift from Paul Anderson lab (PMID:27022092) | | |
| Cell line (*H. sapiens*) | Human osteosarcoma U-2 OS, U-2 OS (ΔΔG3BP1/2) | Gift from Paul Anderson lab (PMID:27022092) | | |
| Cell line (*H. sapiens*) | HeLa | ATCC | Cat # HeLa CCL-2; RRID:CVCL_0030 | |
| Cell line (*Mus musculus*) | Mouse embryonic fibroblasts, NIH/3T3 | ATCC | Cat # CRL-1658; RRID:CVCL_0594 | |
| Chemical compound, drug | DMSO | MilliporeSigma | D8418 | |
| Chemical compound, drug | NaAsO₂ | MilliporeSigma | Cat # S7400-100G | |
| Chemical compound, drug | H2O2 | MilliporeSigma | | |
| Chemical compound, drug | ISRIB | MilliporeSigma | Cat # SML0843-5MG | |
| Chemical compound, drug | CHX | MilliporeSigma | Cat # 239765-1ML | |
| Chemical compound, drug | PMSF | Roche Diagnostics | Cat # 10837091001 | |
| Chemical compound, drug | Protease inhibitor cocktail | Roche Diagnostics | Cat # 11836153001 | |
| Chemical compound, drug | Chemiluminescence | Azure Biosystems | Cat # AC2204 | |
| Chemical compound, drug | Phosphatase inhibitors | MilliporeSigma | Cat # Cocktail 2P5726, Cocktail 3-P0044 | |
| Chemical compound, drug | HALT Protease | Thermo Fisher | Cat # 78740 | |
| Chemical compound, drug | TRIzol reagent | Invitrogen | Cat # 15596-018 | |
| Chemical compound, drug | Bovine serum albumin | MilliporeSigma | Cat # 10735078001 | |
| Chemical compound, drug | Fetal bovine serum | GeminiBio | Cat # 100-106 | |

*Continued on next page*

*Continued*

| Reagent type (species) or resource | Designation | Source or reference | Identifiers | Additional information |
|---|---|---|---|---|
| Chemical compound, drug | ʟ-Glutamine | Thermo Fisher | Cat # 25030081 | |
| Chemical compound, drug | MEM-nonessential amino acids | Invitrogen | Cat # 11140050 | |
| Chemical compound, drug | 4,6-Diamidino-2-phenylindole (DAPI) | MilliporeSigma | Cat # D8417-1MG | |
| Chemical compound, drug | Dulbecco's Modified Eagle Medium, DMEM | Thermo Fisher Scientific | Cat # 11965-092 | |
| Commercial assay or kit | Qubit protein assay | Invitrogen | Cat # Q33211 | |
| Commercial assay or kit | MTS Assay, CellTiter 96 AQueous One Solution Cell Proliferation Assay | Promega | Cat # G3582 | |
| Commercial assay or kit | High Sensitivity (HS) RNA Qubit assay | Invitrogen | Cat # Q32852 | |
| Commercial assay or kit | Qubit 1X dsDNA High Sensitivity (HS) assay kit | Thermo Fisher | Cat # Q33231 | |
| Commercial assay or kit | Qubit RNA IQ assay | Thermo Fisher | Cat # Q33222 | |
| Commercial assay or kit | High Sensitivity DNA kit | Agilent Technologies | 5067-4626 | |
| Commercial assay or kit | Universal mycoplasma detection kit | ATCC | Cat # 30-1012K | |
| Commercial assay or kit | Direct RNA sequencing kit | Oxford Nanopore Technologies | SQK-RNA002 | |
| Commercial assay or kit | Oligo d(T)25 Magnetic Beads | New England Biolabs | Cat # s1419S | |
| Commercial assay or kit | T4 RNA ligase | New England Biolabs | Cat # M0204S | |
| Commercial assay or kit | Direct RNA sequencing Flow Cells (MinION) | Oxford Nanopore Technologies | FLO-MIN106 | |
| Commercial assay or kit | Direct RNA sequencing Flow Cells (PromethION) | Oxford Nanopore Technologies | FLO-PRO002 | |
| Commercial assay or kit | Reverse Transcriptase Superscript III First-Strand Synthesis System | Invitrogen | Cat # 18080-051 | |
| Commercial assay or kit | FastStart SYBR Green Master Mix | KAPA Biosystems | Cat # KK4605/07959435001 | |
| Commercial assay or kit | NuPAGE 4–12% Bis-Tris Gel | Invitrogen | Cat # NP0321BOX | |
| Commercial assay or kit | Polyvinylidene fluoride membrane (PVDF) | Millipore | Cat# IPVH00010 | |
| Commercial assay or kit | ProLong Glass Antifade Mountant | Invitrogen | Cat # P36982 | |
| Recombinant DNA reagent | pT7-EGFP-C1-HsNot8-D40AE42A_AH (Plasmid) | Gift from Elisa Izaurralde | Plasmid # 148902; RRID:Addgene_148902 | |
| Recombinant DNA reagent | pT7-EGFP-C1-HsDCP2-E148Q_U (Plasmid) | Gift from Elisa Izaurralde | Plasmid # 147650; RRID:Addgene_147650 | |
| Recombinant DNA reagent | pEGFP-C1 (Plasmid) | Gift from Myriam Gorospe | | Available upon request |
| Sequence-based reagent | Linker-REL5 (Oligo) | IDT | PMID:34428294 | (/5PCBio/rArArUrGrArUrAr CrGrGrCrGrArCrCrArCrCrGr ArGrArUrCrUrArCrArCrUrCr UrUrUrCrCrCrUrArCrArCrGr ArCrGrCrUrCrUrUrCrCrGrArUrCrU) |
| Sequence-based reagent | siRNA: ON-TARGETplus Set of 4 siRNA J-013754-09, XRN1 #1 | Dharmacon | Cat # J-013754-09-0005 | CUUCAUAGUUGGUCGGUAU |

*Continued on next page*

*Continued*

| Reagent type (species) or resource | Designation | Source or reference | Identifiers | Additional information |
|---|---|---|---|---|
| Sequence-based reagent | siRNA: siGENOME Non-Targeting siRNA #3 | Dharmacon | Cat # D-001210-03-20 | AUGUAUUGGCCUGUAUUAG |
| Software, algorithm | Guppy | https://nanoporetech.com/document/Guppy-protocol | Guppy (3.4.5); RRID:SCR_023196 | |
| Software, algorithm | Cutadapt | DOI: 10.14806/ej.17.1.200 | Cutadapt (2.8); RRID:SCR_011841 | |
| Software, algorithm | Minimap2 | PMID:29750242 | Minimap2 (2.17); RRID:SCR_018550 | |
| Software, algorithm | DESeq2 | PMID:31740818 | DESeq2; RRID:SCR_015687 | |
| Software, algorithm | Nanopolish | https://github.com/jts/nanopolish | Nanopolish (0.14.0); RRID:SCR_016157 | |
| Software, algorithm | ClusterProfiler | PMID:22455463 | ClusterProfiler (4.0); RRID:SCR_016884 | |
| Software, algorithm | STAR | PMID:23104886 | STAR (2.5.3a); RRID:SCR_004463 | |
| Software, algorithm | Pysam | https://github.com/pysam-developers/pysam | Pysam (v0.15.4); RRID:SCR_021017 | |
| Software, algorithm | NanopLen | This paper, https://github.com/maragkakislab/nanoplen | NanopLen | NanopLen is available open source under NIA Public Domain license |
| Software, algorithm | EnhancedVolcano | https://github.com/kevinblighe/EnhancedVolcano | EnhancedVolcano (1.8.0); RRID:SCR_018931 | |
| Software, algorithm | IGV | PMID:36562559 | IGV (2.12.3); RRID:SCR_011793 | |

## Cell lines, cell treatments, and RNA isolation

HeLa (ATCC CCL-2) and U-2 OS (gift from Paul Anderson lab, described in *Kedersha et al., 2016*) cells were cultured at 37°C, 5% $CO_2$, 90% humidity in Dulbecco's Modified Eagle Medium (Thermo Fisher Scientific, Cat# 11965-092) supplemented with 10% heat-inactivated fetal bovine serum (GeminiBio #100-106), 2 mM L-glutamine. NIH3T3 cells (ATCC CRL-1658) were cultured at 37°C, 5% $CO_2$, 90% humidity in Dulbecco's Modified Eagle Medium supplemented with 10% bovine calf serum (GeminiBio), 1% MEM-nonessential amino acids (Invitrogen), 2 mM L-glutamine. The cells were tested for mycoplasma contamination using universal mycoplasma detection kit (ATCC, Cat# 30-1012K). Cells were treated with 500 µM of sodium arsenite (Sigma, Cat# S7400-100G) for 60 min, and total RNA was isolated using TRIzol reagent (Invitrogen, Cat# 15596-018) following the manufacturer's recommendation. RNA concentration was quantified using a High Sensitivity (HS) RNA Qubit assay (Invitrogen, Cat# Q32852) and Nanodrop ND-1000 (Thermo Fisher). RNA integrity was assessed on a 2100 Bioanalyzer (Agilent Technologies) or Qubit RNA IQ assay (Thermo Fisher).

## siRNA and plasmid transfection

Cells were transfected with control and gene-specific siRNAs at a final concentration of 20 nM using the Neon transfection system 100 µl kit (Thermo Fisher, MPK100) according to the manufacturer's protocol. A commercially available siRNA specifically targeting the human XRN1 was used for XRN1 knockdown, and a nontargeting siRNA was used as control (*Supplementary file 2*). Approximately $5 \times 10^5$ cells were resuspended in 100 µl of siRNA-R buffer mixture, electroporated (1005 V, 35 ms, 2 pulses) and immediately transferred to a 100-mm dish containing pre-warmed media. Cells were grown for 72 hr before treatment with indicated reagents and final harvesting for RNA and protein assays.

pT7-EGFP-C1-HsNot8-D40AE42A_AH (Addgene plasmid # 148902; RRID:Addgene_148902) and pT7-EGFP-C1-HsDCP2-E148Q_U (Addgene plasmid # 147650; RRID:Addgene_147650) were a gift from Elisa Izaurralde. Approximately $7.5 \times 10^5$ cells were transfected with 3 µg of these plasmids or the pEGFP-C1 control using the Neon transfection system 10 µl kit (Thermo Fisher, MPK1096, 1005 V, 35 ms, 2 pulses). 24–48 hr post-transfection, cells were treated with 500 µM of sodium arsenite for 1 hr and harvested for RNA and protein extraction.

## Direct RNA sequencing

Library preparation was performed using the direct RNA sequencing kit (Oxford Nanopore Technologies, SQK-RNA002) as previously described (*Ibrahim et al., 2021*) with modifications. A minimum of 75 µg of total RNA was used as starting material to purify poly(A) RNAs using Oligo d(T)25 Magnetic Beads (NEB, Cat# s1419S). 50 pmoles of linker (REL5) containing a 5'-Biotin-PC group and a 3'-OH (*Supplementary file 2*) were ligated to the 5' end of RNAs using T4 RNA ligase (NEB, Cat# M0204S) for 3 hr at 37°C, as previously described (*Ibrahim et al., 2021*). 500–1000 ng of poly(A) RNA was used for library preparation using SQK-RNA002 sequencing kit (Oxford Nanopore Technologies). The final library was quantified using Qubit 1X dsDNA HS assay kit (Thermo Fisher # Q33231) and loaded on FLO-MIN106 or FLO-PRO002 flow cells.

## RT-qPCR

Superscript III (Invitrogen, Cat# 18080-051) was used to synthesize complementary DNA (cDNA). Briefly, 1 µg of total RNA was reverse-transcribed to cDNA using SuperScript III First-Strand Synthesis System (Thermo Fisher) and oligo-dT primers. One-tenth dilution of the cDNA mixture was used to perform qPCR using FastStart SYBR Green Master Mix (KAPA Biosystems, Cat# KK4605/07959435001) and run on a QuantStudio3 thermal cycler (Applied Biosystems, Cat# A28567). Primers are listed in *Supplementary file 2*.

## MTS assay

MTS assay was performed using CellTiter 96 AQueous One Solution Cell Proliferation Assay (MTS) (Promega, Cat# G3582). HeLa cells ($5 \times 10^3$ cells per well) were seeded in a 96-well plate and incubated in humidified 5% $CO_2$ incubator at 37°C for 24 hr. Following arsenite treatment, the cell viability was examined using MTS assay (100 µl of DMEM and 20 µl MTS incubated for indicated time points) for 10 min and incubated in a 5% $CO_2$ incubator at 37°C protected from light. Finally, the plate was subjected to shaking for 15 min followed by an optical density measurement (OD) at 590 nm using 1420 Multilabel Counter (PerkinElmer, VICTOR$^3$V). The viable cells in arsenite-treated samples were reported as the percentage of viable cells in control (untreated) samples.

## Immunoblotting

Cells were washed with cold PBS and pelleted by centrifugation 5 min at 4°C 300 × *g*. Whole-cell lysates were prepared by adding 1 volume of SDS lysis buffer (60 mM Tris-HCl pH 7.5, 2% SDS, 10% glycerol) supplemented with 1× protease inhibitor cocktail (Roche Diagnostics, Cat# 11836153001) and 1 mM phenylmethylsulfonyl fluoride (PMSF) (Roche, Cat# 10837091001). The lysate was passed 10 times through a 25-gauge needle, heated at 95°C for 20 min, and cleared by centrifugation at 16,500 × *g* for 10 min. Protein concentrations were measured using the Qubit protein assay (Invitrogen, Cat# Q33211). 25 µg of proteins was separated on NuPAGE 4–12% Bis-Tris Gel (Invitrogen, Cat# NP0321BOX) and transferred to PVDF membrane (Millipore, Cat# IPVH00010). After blocking in TBS-T 5% milk for 2 hr, membranes were incubated with primary antibodies overnight at 4°C. Membranes were washed three times 5 min in TBS-T and incubated for 2 hr with secondary antibodies. Signals were developed using Chemiluminescence (Azure Biosystems, Cat# AC2204) and acquired on a ChemiDoc MP imaging system (Bio-Rad). For studies of eIF2α phosphorylation (eIF2α-P), the lysis buffer was supplemented with phosphatase inhibitors (Sigma, Cat# Cocktail 2-P5726 and Cocktail 3-P0044) and 5% BSA was used instead of milk for blocking. Antibodies are listed in *Supplementary file 2*.

## Immunofluorescence

Cells were cultured at 70–80% confluency in Millicell EZ SLIDE 8-well glass (Millipore, Cat# PEZGS0816) in the presence of 50, 100, 250, and 500 µM of sodium arsenite ±200 nM of ISRIB (Sigma, Cat# SML0843). After the indicated time of treatment, cells were washed three times with warm PBS, fixed for 10 min at room temperature (RT) in PBS/3.7% paraformaldehyde and washed with PBS for 5 min with shaking. After permeabilization in PBS 0.5% Triton X-100 for 10 min at RT, cells were blocked in PBS 0.1% Tween-20 (PBS-T) supplemented with 1% BSA for 1 hr at 37°C with gentle shaking. Primary antibody diluted in 1% BSA was added to the cells and incubated at 4°C overnight. Cells were washed in PBS-T three times 5 min with shaking before staining with secondary antibody for

1 hr at RT. Cells were washed four times for 5 min with PBS-T with shaking, and nuclei were stained with 4,6-diamidino-2-phenylindole (DAPI) (1:1000 in PBS-T) for 5 min at RT. Slides were mounted with ProLong Glass Antifade Mountant (Invitrogen, Cat# P36982), and images were acquired using a DeltaVision Microscope System (Applied Precision).

## Polysome profiling

The polysome profiling was performed as described previously (*Panda et al., 2017*) with modifications. Briefly, HeLa cells (80–85% confluency) cultured in 100 mm dishes were treated with 500 µM sodium arsenite, 200 nM ISRIB, or 25 µg/ml cycloheximide as indicated. DMSO was used as a control. Immediately before harvesting, cells were treated with 100 µg/ml cycloheximide for 10 min. After one wash with ice-cold PBS containing 100 µg/ml cycloheximide, 500 µl of polysome extraction buffer (20 mM Tris-HCl pH 7.5, 50 mM NaCl, 50 mM KCl, 5 mM MgCl$_2$, 1 mM DTT, 1X HALT Protease, 1.0% Triton X-100, and 100 µg/ml cycloheximide) was added directly to the plate, and lysates were harvested by scraping. The lysate was cleared by centrifugation at 14,000 × *g* for 10 min. The supernatant was loaded onto the 10–50% sucrose gradient followed by high-speed centrifugation (260,800 × *g* for 90 min at 4°C). Using a density gradient fractionation system monitored by UV absorbance detector (A254), 12 fractions were collected.

## Nanopore sequencing data processing

Nanopore sequencing data were basecalled using Guppy (v3.4.5). The 5′ adapters were identified and removed using cutadapt (v2.8). Reads were first aligned against ribosomal sequences obtained from SILVA (*Quast et al., 2013*). Nonribosomal reads were subsequently mapped against the human genome hg38 using minimap2 (version 2.17) (*Li, 2018*) and parameters -a -x splice -k 12 -u b -p 1 `--secondary=yes`. They were also aligned against the human transcriptome using -a -x map-ont -k 12 -u f -p 1 `--secondary=yes`.

## Poly(A) tail length estimation

The poly(A) tail lengths were extracted from sequenced reads using the nanopolish polya package (*Workman et al., 2019*). Only the poly(A) tail lengths that passed the software quality control scores and were tagged as 'PASS' were used in our analysis.

## Differential gene expression and gene ontology

Transcript counts were quantified from the transcriptome alignments as the total number of reads aligning on each transcript. Differential expression analysis was performed using DESeq2 (*Love et al., 2014*). EnhancedVolcano (RRID:SCR_018931) was used for visualization. Gene ontology analysis was performed for significantly changed genes (p-value<0.05) using 'clusterProfiler' (*Yu et al., 2012*).

## Ribosome profiling data

Ribosome profiling sequencing data were downloaded from GEO:GSE79664 (*Park et al., 2016*). Adapters were removed using cutadapt (version 2.8), and reads were aligned to the human genome using STAR (2.5.3a). Further processing to calculate counts per transcript was performed with in-house scripts using pysam (v0.15.4), and counts were converted to reads per kilobase of transcript per million.

## Meta-length calculation

For calculating the bin (meta) lengths, we divided the transcripts into 20 equal bins [0, 19]. Reads were assigned to each of these bins based on the corresponding location of their ends. The binned- (meta-) length is calculated as the difference of the binned 3′ end over the binned 5′ end.

## TSS identification

TSS identification was performed as previously described (*Ibrahim et al., 2021*). Briefly, all reads from all replicates were combined and the distribution of read 5′ ends within the 5′ UTR was quantified. The TSS was selected as the position with the highest read density withing the 5′ UTR with minimum five supporting reads.

## NanopLen

NanopLen reads a file of read lengths with library identifiers and gene or transcript identifiers, and a metadata file describing the experimental design. The latter includes the library identifiers and the corresponding condition for each library. The software supports three models: *t*-test, Wilcoxon, and linear mixed model (LMM). In the *t*-test and LMM options, the user can also supply a customized model to use extra variables in the metadata. In this work we have used the LMM to adjust for putative batch effects across libraries.

Given a vector Y of read lengths associated with a gene/transcript, the *t*-test is functionally equivalent to a linear regression model. The model also supports optional read-specific extra covariates but are not used in this work:

$$Y = \beta_0 + \beta_{cond}cond + \epsilon + \left(\text{extra covariates}\right).$$

Similarly, the LMM models Y using the condition as fixed effect and the library identifier as a random effect.

$$Y = \beta_0 + \beta_{cond}cond + lib + \epsilon + \left(\text{extra covariates}\right),$$

where *lib* is the random effect of each library. By adding the random effect, the analysis is more robust against potential false positives from library variation and prevents deeper libraries from dominating the effect size. A Wald test is used to test for significance of $\beta_{cond}$. The Wilcoxon test option is restricted to only testing the condition variable and cannot adjust for additional variables. Each gene/transcript is assigned a statistic corresponding to the test used and a p-value. p-Values are adjusted for multiple testing using the Bonferroni–Hochberg method.

To test NanopLen models, transcript length data with known shortening rates were simulated. Each simulation was parameterized with a 'true' transcript length, a read count as a proxy for transcript expression, and a shortening proportion. The database of known human genic lengths from Ensembl (release 91) was used and the median was selected as 'true' length. Since the simulated variance is in proportion of the true length, using any length will result in similar results, so we do not simulate additional true lengths. Expression counts were simulated, ranging from 10 to 200 reads per transcripts to capture the dynamic expression range particularly toward small read depth. Varying shortening percentages were simulated from 50 to 100% of the true length, with the 100% corresponding to the null simulation of no-change.

Given the parameters, the simulated lengths were generated based on the mixed model in NanopLen, with the number of sampled lengths being the selected expression counts. The library random effect was simulated as if imitating a fluctuation with standard deviation of 10% of the true length, and the global error having a standard deviation of 20% of the expected length mean. As with the real experiment, six libraries of three control and three condition were simulated, with 1000 genes per scenario. Using the LMM option with the logscale option selected, the resulting log2FCs and p-values were calculated.

NanopLen is available open source under NIA Public Domain license on GitHub, (copy archived at *Dar et al., 2024*).

## IGV read density plots

Transcripts that were identified as significantly shortened at a false discovery rate of 0.05 by NanopLen, had at least 50 supporting reads in both conditions, and had a difference of 200 nt in length were selected. From the list of 32 transcripts, 3 were randomly selected (*ASCC3*, *HNRNPM,* and *DSG2*) for visualization. Libraries were randomly downsampled to maximum 50 reads per window, and the libraries were overlayed in Affinity Designer. All reads were used, irrespective of adapter ligation status.

## Acknowledgements

We thank Dr. Paul Anderson for his gift of the ΔΔG3BP1/2 and WT U-2 OS cells. We thank Dr. David Schlessinger for providing feedback on our manuscript. We thank the NIH Fellows Editorial Board for editorial assistance. This research was supported by the Intramural Research Program of the National Institute on Aging, National Institutes of Health and funded by grant NIH ZIA AG000696 to MM.

# Additional information

## Funding

| Funder | Grant reference number | Author |
|---|---|---|
| National Institute on Aging | ZIA AG000696 | Manolis Maragkakis |

The funders had no role in study design, data collection and interpretation, or the decision to submit the work for publication.

## Author contributions

Showkat Ahmad Dar, Data curation, Software, Formal analysis, Investigation, Visualization, Methodology, Writing – original draft, Project administration; Sulochan Malla, Conceptualization, Formal analysis, Validation, Investigation, Visualization, Methodology, Writing – original draft, Project administration; Vlastimil Martinek, Formal analysis, Visualization; Matthew John Payea, Investigation, Writing – review and editing; Christopher Tai-Yi Lee, Software, Formal analysis, Methodology; Jessica Martin, Aditya Jignesh Khandeshi, Jennifer L Martindale, Formal analysis; Cedric Belair, Validation, Investigation, Project administration, Writing – review and editing; Manolis Maragkakis, Conceptualization, Resources, Data curation, Software, Formal analysis, Supervision, Funding acquisition, Visualization, Methodology, Writing – original draft, Project administration, Writing – review and editing

## Author ORCIDs

Showkat Ahmad Dar ⬦ https://orcid.org/0000-0002-5077-1925
Sulochan Malla ⬦ https://orcid.org/0000-0001-9957-4597
Vlastimil Martinek ⬦ https://orcid.org/0000-0002-3204-1830
Matthew John Payea ⬦ https://orcid.org/0000-0002-1960-4563
Christopher Tai-Yi Lee ⬦ https://orcid.org/0000-0002-8621-256X
Jessica Martin ⬦ https://orcid.org/0009-0000-0830-9289
Aditya Jignesh Khandeshi ⬦ http://orcid.org/0009-0004-8541-1702
Jennifer L Martindale ⬦ http://orcid.org/0000-0002-3234-6861
Cedric Belair ⬦ https://orcid.org/0000-0003-4007-2060
Manolis Maragkakis ⬦ https://orcid.org/0000-0002-3158-1763

Reviewer #1 (Public Review): https://doi.org/10.7554/eLife.96284.3.sa1
Reviewer #2 (Public Review): https://doi.org/10.7554/eLife.96284.3.sa2
Reviewer #3 (Public Review): https://doi.org/10.7554/eLife.96284.3.sa3
Author response https://doi.org/10.7554/eLife.96284.3.sa4

# Additional files

## Supplementary files

- Supplementary file 1. Sequencing library list and metadata.
- Supplementary file 2. RT-qPCR primers, siRNAs, antibodies, and oligos.
- Supplementary file 3. Differential expression analysis for arsenite-treated vs. control libraries.
- Supplementary file 4. Differential length analysis for arsenite-treated vs. control cells.
- MDAR checklist

## Data availability

Sequencing data have been deposited in the Gene Expression Omnibus (GEO); accession: GSE204785. Source code for NanopLen is available in GitHub, (copy archived at *Dar et al., 2024*).

The following dataset was generated:

| Author(s) | Year | Dataset title | Dataset URL | Database and Identifier |
|---|---|---|---|---|
| Dar SA, Malla S, Martinek V, Payea MJ, Lee CT, Martin J, Khandeshi AJ, Martindale JL, Belair C, Maragkakis M | 2024 | Full-length direct RNA sequencing uncovers stress-granule dependent RNA decay upon cellular stress | https://www.ncbi.nlm.nih.gov/geo/query/acc.cgi?acc=GSE204785 | NCBI Gene Expression Omnibus, GSE204785 |

The following previously published datasets were used:

| Author(s) | Year | Dataset title | Dataset URL | Database and Identifier |
|---|---|---|---|---|
| Park J, Yi H, Kim Y, Chang H, Kim VN | 2016 | Regulation of poly(A) tail and translation during the somatic cell cycle | https://www.ncbi.nlm.nih.gov/geo/query/acc.cgi?acc=GSE79664 | NCBI Gene Expression Omnibus, GSE79664 |
| Maier K, Gressel S, Cramer P, Schwalb B | 2020 | Native molecule sequencing by nano-ID reveals synthesis and stability of RNA isoforms | https://www.ncbi.nlm.nih.gov/geo/query/acc.cgi?acc=GSE127890 | NCBI Gene Expression Omnibus, GSE127890 |
| Watkins CP, Zhang W, Wylder A, Katanski CD, Pan T | 2022 | A multiplex platform for small RNA sequencing elucidates multifaceted tRNA stress response and translational regulation | https://www.ncbi.nlm.nih.gov/geo/query/acc.cgi?acc=GSE198441 | NCBI Gene Expression Omnibus, GSE198441 |

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
