## [Editor Report · eLife Assessment]

This **important** study describes mRNA shortening during cellular stress and interestingly observes that this shortening is dependent on localization in stress granules. Surprisingly, this mRNA shortening does not appear to require the shortening of poly A tails. These are novel, paradigm-shifting findings, using cutting-edge technologies and **convincing** data, that should be of broad interest to the RNA community and beyond.

---

## [Referee Report · Reviewer #1 (Public Review)]

In this manuscript, the authors employed direct RNA sequencing with nanopores, enhanced by 5' end adaptor ligation, to comprehensively interrogate the human transcriptome at single-molecule and nucleotide resolution. They conclude that cellular stress induces prevalent 5' end RNA decay that is coupled to translation and ribosome occupancy. Contrary to the literature, they found that, unlike typical RNA decay models in normal conditions, stress-induced RNA decay is dependent on XRN1 but does not depend on the removal of the poly(A) tail. The findings presented are interesting and the authors fully established these paradigm-shifting findings using cutting-edge technologies.

---

## [Referee Report · Reviewer #2 (Public Review)]

In the manuscript "Full-length direct RNA sequencing uncovers stress-granule dependent RNA decay upon cellular stress", Dar, Malla, and colleagues use direct RNA sequencing on nanopores to characterize the transcriptome after arsenite and oxidative stress. They observe a population of transcripts that are shortened during stress. The authors hypothesize that this shortening is mediated by the 5'-3' exonuclease XRN1, as XRN1 knockdown results in longer transcripts. Interestingly, the authors do not observe a polyA-tail shortening, which is typically thought to precede decapping and XRN1-mediated transcript decay. Finally, the authors use G3BP1 knockout cells to demonstrate that stress granule formation is required for the observed transcript shortening. The manuscript contains intriguing findings of interest to the mRNA decay community.

---

## [Referee Report · Reviewer #3 (Public Review)]

The work by Dar et al. examines RNA metabolism under cellular stress, focusing on stress-granule-dependent RNA decay. It employs direct RNA sequencing with a Nanopore-based method, revealing that cellular stress induces prevalent 5' end RNA decay that is coupled to translation and ribosome occupancy but is independent of the shortening of the poly(A) tail. This decay, however, is dependent on XRN1 and enriched in the stress granule transcriptome. Notably, inhibiting stress granule formation in G3BP1/2-null cells restores the RNA length to the same level as wild-type. It suppresses stress-induced decay, identifying RNA decay as a critical determinant of RNA metabolism during cellular stress and highlighting its dependence on stress-granule formation. This is an exciting and novel discovery utilizing innovative sequencing methods to studying mRNA decay.

---

## [Author Response]

The following is the authors’ response to the original reviews.

**Public Reviews:**

**Reviewer #1 (Public Review):**
Summary:In this manuscript, the authors employed direct RNA sequencing with nanopores, enhanced by 5' end adaptor ligation, to comprehensively interrogate the human transcriptome at singlemolecule and nucleotide resolution. They conclude that cellular stress induces prevalent 5' end RNA decay that is coupled to translation and ribosome occupancy. Contrary to the literature, they found that, unlike typical RNA decay models in normal conditions, stress-induced RNA decay is dependent on XRN1 but does not depend on the removal of the poly(A) tail. The findings presented are interesting but a substantial amount of work is needed to fully establish these paradigm-shifting findings.Strengths:These are paradigm-shifting observations using cutting-edge technologies.Weaknesses:The conclusions do not appear to be fully supported by the data presented.

Our response to the reviewer comments is provided at the end of this document in the section "Recommendations For The Authors"

**Reviewer #2 (Public Review):**
In the manuscript "Full-length direct RNA sequencing uncovers stress-granule dependent RNA decay upon cellular stress", Dar, Malla, and colleagues use direct RNA sequencing on nanopores to characterize the transcriptome after arsenite and oxidative stress. They observe a population of transcripts that are shortened during stress. The authors hypothesize that this shortening is mediated by the 5'-3' exonuclease XRN1, as XRN1 knockdown results in longer transcripts. Interestingly, the authors do not observe a polyA-tail shortening, which is typically thought to precede decapping and XRN1-mediated transcript decay. Finally, the authors use G3BP1 knockout cells to demonstrate that stress granule formation is required for the observed transcript shortening.The manuscript contains intriguing findings of interest to the mRNA decay community. That said, it appears that the authors at times overinterpret the data they get from a handful of direct RNA sequencing experiments. To bolster some of the statements additional experiments might be desirable.A selection of comments:(1) Considering that the authors compare the effects of stress, stress granule formation, and XRN1 loss on transcriptome profiles, it would be desirable to use a single-cell system (and validated in a few more). Most of the direct RNAseq is performed in HeLa cells, but the experiments showing that stress granule formation is required come from U2OS cells, while short RNAseq data showing loss of coverage on mRNA 5'ends is reanalyzed from HEK293 cells. It may be plausible that the same pathways operate in all those cells, but it is not rigorously demonstrated.

We agree with the reviewer that performing all experiments in a single cell system would be desirable. Presently, our core findings on 5’ RNA shortening are all performed in HeLa cells: the identification of 5’ RNA shortening, the reliance of shortening through XRN1 silencing, suppression of shortening by translation inhibition, and now the relationship between 5’ shortening and deadenylation/decapping through experiments described further below. Our use of other cell lines is primarily to show that 5’ shortening is a general phenomenon, and we have now done this for U20S cells, HEK293 cells, and primary 3T3 cells from mouse.

Regarding stress granule formation, we are unfortunately restricted by the lack of available well characterized resources. The DDG3BP1/2 U2OS is a well characterized cell line that has been extensively used for stress granule-related experiments. We have therefore opted to use it and performed experiments to verify both the occurrence of stress-induced RNA shortening as well as the rescue in the absence of stress granules. The reproducibility and breadth of the cell lines used in our analysis makes us confident on the generality of our findings.

(2) An interesting finding of the manuscript is that polyA tail shortening is not observed prior to transcript shortening. The authors would need to demonstrate that their approach is capable of detecting shortened polyA tails. Using polyA purified RNA to look at the status of polyA tail length may not be ideal (as avidity to oligodT beads may increase with polyA tail length and therefore the authors bias themselves to longer tails anyway). At the very least, the use of positive controls would be desirable; e.g. knockdown of CCR4/NOT.

We thank the reviewer for their comment. Previous studies, using in vitro transcribed RNA molecules, have shown that direct RNA sequencing can capture and quantify poly(A) tails of varying lengths (Krause et al. 2019). Specifically, a range of 10 to 150 nt has been tested and a high concordance between known and dRNA-Seq determined values was observed. Both tailfindR and nanopolish (used in this work) showed high poly(A) tail estimation accuracy.

Regardless, we agree with the reviewer that our method depends on poly(A) tail capture and thus may be incomplete for fully quantifying poly(A) length changes. We therefore opted to replace these data and instead follow this and other reviewers’ suggestions and perform experiments following knockdown of CCR4/NOT using cells expressing a catalytically inactive CNOT8 (CNOT8*) dominant negative mutant (Chang et al. 2019). Our new data show that stress-induced 5’ end decay is indeed not dependent on prior removal of the poly(A) tail. Specifically, we find that transcript shortening is still observed upon oxidative stress in cells expressing CNOT8* compared to control cells. We present these new results in Fig. 3 and Sup. Fig 3.

(3) The authors use a strategy of ligating an adapter to 5' phosphorylated RNA (presumably the breakdown fragments) to be able to distinguish true mRNA fragments from artifacts of abortive nanopore sequencing. This is a fantastic approach to curating a clean dataset. Unfortunately, the authors don't appear to go through with discarding fragments that are not adapter-ligated (presumably to increase the depth of analysis; they do offer Figure 1e that shows similar changes in transcript length for fragments with adapter, compared to Figure 1d). It would be good to know how many reads in total had the adapter. Furthermore, it would be good to know what percentage of reads without adapters are products of abortive sequencing. What percentage of reads had 5'OH ends (could be answered by ligating a different adapter to kinasetreated transcripts). More read curation would also be desirable when building the metagene analysis - why do the authors include every 3'end of sequenced reads (their RNA purification scheme requires a polyA tail, so non-polyadenylated fragments are recovered in a nonquantitative manner and should be discarded).

We thank the reviewer for appreciating our approach. The reviewer is correct that we do not discard reads that are not adapter-ligated. As the reviewer correctly mentions this is to increase the sequencing depth. We have found that the ligation efficiency is very low, ~1-2 % of total reads (now in Sup. Table. 1), across all libraries, and so the percentage of REL5-ligated reads does not directly infer the total amount of non-artifactual 5’ ends. Instead, we use these REL5-ligated reads as a subset of our data for which we have extremely high confidence in the true 5’end. Our results show that non-ligated reads display the same length distribution as ligated ones, and that the results are reproducible regardless of read selection (e.g. Fig. 1c, e, Sup. Fig. 1k, l, Fig. 3b, c). This strong concordance between REL5-ligated and non-ligated reads suggests that our conclusions on 5’ end shortening are not substantially influenced by abortive sequencing or other artefactual creation of 5’ shortening. We have modified the text to clarify these points and have added plots using only ligated molecules for relevant figures that this was not previously done (Sup. Fig 1l, 3c)

We agree with the reviewer that non-polyadenylated reads could be discarded from metagene analysis and we have performed this change in the revised version. Our conclusions following removal of non-polyadenylated reads remain unchanged (Sup. Fig. 1g).

(4) The authors should come to a clear conclusion about what "transcript shortening" means. Is it exonucleolytic shortening from the 5'end? They cannot say much about the 3'ends anyway (see above). Or are we talking about endonucleolytic cuts leaving 5'P that then can be attached by XRN1 (again, what is the ratio of 5'P and 5'OH fragments; also, what is the ratio of shortened to full-length RNA)?

We thank the reviewer for their suggestion. We have performed additional experiments to investigate the role of deadenylation and decapping by expressing dominant negative forms of the NOT8 deadenylase (NOT8*) and DCP2 decapping (DCP2*) enzyme in HeLa cells. Our results show that neither expression of NOT8* nor DCP2* can inhibit stress-induced transcript shortening following arsenite treatment (Fig. 3e-f). These new data suggest that neither deadenylation nor decapping are required for stress-induced RNA decay. Instead, our data are more compatible with endonucleolytic cleavage as the most likely mechanism for stressinduced RNA decay. We have incorporated these results in the text and present them in Fig. 3 and Sup. Fig. 3.

(5) The authors should clearly explain how they think the transcript shortening comes about. They claim it does not need polyA shortening, but then do not explain where the XRN1 substrate comes from. Does their effect require decapping? Or endonucleolytic attacks?

Please also refer to our answer to the previous comment (#4). Collectively, our results from (a) the dominant negative expression of NOT8* and DCP2* that show no effect on stress-induced shortening and (b) the rescue of transcript length upon translation initiation inhibition, indicate a potential endonucleolytic mechanism as a mediator of stress-induced RNA decay. However, we believe that extensive, further studies currently beyond the scope of this work, will be required to discover the nuclease and to dissect the exact molecular mechanisms that define the 5' ends of mRNAs upon stress-induced decay. We now discuss these points in the discussion.

(6) XRN1 KD results in lengthened transcripts. That is not surprising as XRN1 is an exonuclease - and XRN1 does not merely rescue arsenite stress-mediated transcript shortening, but results in a dramatic transcript lengthening.

The reviewer raises an intriguing point. Additional analysis of data has showed that in fact, in unstressed cells, XRN1 KD leads to modestly significant reduction in overall transcript length (Fig. 3b, c). This could possibly be the result of an accumulation of intermediate cleavage products normally expected to be degraded by XRN1 as previously described (Pelechano, Wei, and Steinmetz 2015; Ibrahim et al. 2018).

Instead, we find that under stress, XRN1 KD shows an almost identical transcript length distribution to unstressed cells and significantly higher than siCTRL stressed cells (Fig. 3b, c). These results indicate that in the absence of XRN1, stress-induced decay is largely abolished. As the reviewer correctly points out, this seems to affect the majority of RNAs which we believe is evidence of the general lack of specificity in the mechanism. Nevertheless, we find that transcripts that are the primary substrates to stress-induced shortening are substantially more lengthened than all other transcripts (Fig. 3e). This indicates that transcripts primarily affected by stress-induced decay are also lengthened the most in the absence of XRN1 and at an even higher level than expected by general XRN1 KD effects.

**Reviewer #3 (Public Review):**
The work by Dar et al. examines RNA metabolism under cellular stress, focusing on stressgranule-dependent RNA decay. It employs direct RNA sequencing with a Nanopore-based method, revealing that cellular stress induces prevalent 5' end RNA decay that is coupled to translation and ribosome occupancy but is independent of the shortening of the poly(A) tail. This decay, however, is dependent on XRN1 and enriched in the stress granule transcriptome. Notably, inhibiting stress granule formation in G3BP1/2-null cells restores the RNA length to the same level as wild-type. It suppresses stress-induced decay, identifying RNA decay as a critical determinant of RNA metabolism during cellular stress and highlighting its dependence on stress-granule formation.This is an exciting and novel discovery. I am not an expert in sequencing technologies or sequencing data analysis, so I will limit my comments purely to biology and not technical points. The PI is a leader in applying innovative sequencing methods to studying mRNA decay.One aspect that appeared overlooked is that poly(A) tail shortening per se does lead to decapping. It is shortening below a certain threshold of 8-10 As that triggers decapping. Therefore, I found the conclusion that poly(A) tail shortening is not required for stress-induced decay to be somewhat premature. For a robust test of this hypothesis, the authors should consider performing their analysis in conditions where CNOT7/8 is knocked down with siRNA.

We agree with the reviewer. We have now performed experiments in cells expressing a well characterized catalytically inactive dominant negative NOT8 isoform (NOT8*) (Chang et al. 2019). Our new data show that stress-induced decay still occurs in cells expressing NOT8*.

These results confirm our findings that stress-induced decay does not require deadenylation. We present these new results in Fig. 3 and Sup. Fig. 3.

Similarly, as XRN1 requires decapping to take place, it necessitates the experiment where a dominant-negative DCP2 mutant is over-expressed.

We agree with the reviewer and have performed this experiment as requested. Expression of a dominant negative DCP2 (DCP2*) isoform (Loh, Jonas, and Izaurralde 2013) in HeLa cells showed that decapping is also not required for stress-induced decay. We present these new results in Fig. 3 and Sup. Fig. 3.

Are G3BP1/2 stress granules required for stress-induced decay or simply sites for storage? This part seems unclear. A very worthwhile test here would be to assess in XRN1-null background.

We thank the reviewer for their comment. Our data show that stress-induced decay is not observed in DDG3BP1/2 U2OS cells, unable to form stress granules (Fig. 6). This result suggests that G3BP1/2 SGs are either (a) required for 5’ RNA shortening or (b) preserve partially fragmented RNAs that would otherwise be rapidly degraded. We find the second option unlikely for two reasons. First, even if the fragments were rapidly degraded, we would still expect to find evidence of their presence in our data. However, Fig. 6f shows that the length distribution of DDG3BP1/2 U2OS cells, with and without arsenite, are almost identical, thus arguing against the presence of such a pool of rapidly degrading RNAs. Second, if these RNAs were protected by SGs, then they would be expected to be downregulated in the absence of SGs in DDG3BP1/2 U2OS cells treated with arsenite. Our results contradict this hypothesis as no association is found between the level of downregulation in arsenite-treated DDG3BP1/2 U2OS cells and the observed stress-induced fragmentation in WT. Collectively our results point towards G3BP1/2 stress granules being required for stress-induced decay. We have expanded on these points in the manuscript to clarify.

Finally, the authors speculate that the mechanism of stress-induced decay may have evolved to relieve translational load during stress. But why degrade the 5' end when removing the cap may be sufficient? This returns to the question of assessing the role of decapping in this mechanism.

The reviewer raises a very interesting point. Our new results, following expression of dominant negative DCP2, show that stress-induced decay does not require decapping. It is therefore plausible that a stress-induced co-translational mechanism cleaves mRNAs endonucleolyticaly to reduce the translational load. Such a mechanism would have many functional benefits as it would acutely reduce the translational load, degrade non-essential RNAs, preserve energy and release ribosomes for translation of the stress response program. We have expanded the discussion to mention these points.

**Recommendations for the authors:**

**Reviewing Editor (Recommendations For The Authors):**
As you can see from the comments, although the reviewers appreciate the novelty of your findings, there was a consensus opinion from all reviewers that the authors overinterpreted their data, since they only have one assay and did not fully analyze it, as laid out in one of the reviewer's critiques. Some orthogonal validation of the "groundbreaking" claims is necessary. Examination of the effects of upstream events in 5'-to-3' decay, namely deadenylation, and decapping, would be necessary for a better understanding of the phenomena the authors describe. Many tools and approaches for studying this are described well in the literature (CNOT7-KD, dominant negative DCP2 E148Q, XRN1-null cell lines), so it is well within the authors' reach. Overall, while some of the evidence presented is novel and solid, for some of the claims there is only incomplete evidence.

We thank the reviewers and the editor for their comments and suggestions. We have performed several additional experiments to further support our conclusions. We have notably investigated the role of deadenylation and decapping in the stress-induced decay by expressing dominant negative NOT8 and DCP2, respectively, as suggested. Our results show that neither deadenylation nor decapping is necessary for stress-induced transcript shortening, suggesting an endonucleolytic event. We believe that these additional experiments strengthen the main conclusions of our work.

**Reviewer #1 (Recommendations For The Authors):**
Major comments:(1) The experiments were conducted in two unrelated cell lines, HeLa and U2OS. The authors should determine if the 5'end RNA decay in response to stress is also observed in normal human cells such as normal human diploid fibroblasts. Furthermore, it would be important to know if this mechanism is conserved between human and mouse cells. This can be tested in mouse embryonic fibroblasts.

We thank the reviewer for their suggestion. We have now also performed experiments in the mouse embryonic fibroblast NIH 3T3 cell line. Our new results confirm that stress-induced 5’ end RNA decay is also observed in this primary cell line and is conserved between human and mouse (Sup. Fig. 1k, I).

(2) The authors state that they monitored cell viability up to 24 hours after Arsenite treatment, but the data is shown up to 240 min (Suppl. 1a). Also, the Y-axis label of this Figure is "Active cells (%)". This should be changed to "Live cells (%)" if this is what they are referring to.

We thank the reviewer for identifying this mistake. Cell viability was monitored up to 4 hours after arsenite treatment. We have corrected the text and modified the figure according to the reviewer’s suggestion.

(3) Based on direct Nanopore-based RNA-seq the authors surprisingly found that RNAs in oxidative stress were globally shorter than unstressed cells. Since Nanopore-based RNA-seq will not detect RNAs that lack a poly A-tail, are they not missing out on RNAs that have already started getting degraded due to the loss of a poly A-tail? Also, I am not sure if they used a spikein control which would be critical to claim global changes in RNA expression.

We agree with the reviewer that our strategy does not capture RNA molecules without a poly(A) tail. Nevertheless, our data do identify shortening upon stress at the 5’ end of RNAs that include poly(A) tails. We considered this as direct evidence that decay at the 5’ end does not require prior removal of the poly(A) tail. Otherwise, these molecules would not have been captured and observed. Indeed, our newly added data from cells expressing a well characterized catalytically inactive dominant negative NOT8 isoform (Chang et al. 2019) show that stress-induced decay occurs even upon silencing of the CCR4-NOT deadenylation complex. We present these results in Fig. 3 and Sup. Fig 3.

We would like to clarify that in our results we did not use a spike-in control and thus refrain from claiming global changes in RNA expression. Instead, we compare relative ratios of groups of molecules within libraries that are internally normalized, we perform correlative comparisons that are invariant to normalization and we perform differential gene expression using established normalization schemes such as DESeq2 (Love, Huber, and Anders 2014).

(4) Many graphs are confusing and inconsistent. For example, samples for Nanopore RNA-seq were prepared in triplicates. Biological or technical? The schematic in Figure 1a shows ISRIB but it appears from Figure 4 onwards. It is missing in the Figure 1 results and the Figure legend. The X-axis labels of many graphs are confusing. For example, Supplementary Figure 1d, 1e, 1g and 1h. It says transcript length but are these nucleotides? P-values are missing from many of these graphs. For some graphs, the authors compared Unstressed vs Arsenite (Figure 1), but in other panels they state No Ars vs 0.5 mM Ars (Fig. 3a) or Control vs Ars (Figure 5c). Likewise, in Figure 1b, Expression change (log2) is unstressed vs Arsenite or Arsenite vs unstressed?

We thank the reviewer identifying these inconsistencies in the presentation of our results. The replicates for nanopore RNA-seq experiments were biological. We have now clarified this point in the text. Furthermore, we have removed “ISRIB” from Fig. 1a to avoid any confusion. We have also made our labelling across all figures more consistent using ‘unstressed’ for NO arsenite treatment vs “arsenite” or ‘+ Ars’ for arsenite treatment.

(5) The authors transfected cells with siCTRL or siXRN1 using electroporation and treated the cells 72 hours after transfection. Since XRN1 is an essential gene, it would be important to determine the viability of cells 72 hours after transfection. Along these lines, in Figure 3b, it would be important to determine the effect of XRN1 knockdown in unstressed cells. Currently, there are only 3 comparisons in Figure 3b - unstressed, siCTRL + Ars and siXRN1 + Ars, and this is insufficient to conclude the effects of XRN1 knockdown in the presence of Arsenite.

We thank the reviewer for their suggestion. We have updated Fig. 3b and the text to show the requested conditions: siCTRL and siXRN1 with and without arsenite. While XRN2 is an essential gene for many organisms, XRN1 is not essential in mammalian cells and no increased cell death has been reported for XRN1-KO or –KD cells (Brothers et al. 2023). We have also tested different concentration (up to 40 nM) of siRNA and monitored the cells up to five days after transfection without observing any cell toxicity, as previously reported.

(6) More broadly, the whole study is somewhat descriptive. The biological effect of 5'end mRNA shortening on gene expression is unclear. There is no data indicating how these changes in RNA lengths impact protein expression. Global quantitative proteomics would be critical to determine this.

We thank the reviewer for their suggestion. To address this concern we have performed additional experiments using cells expressing catalytically inactive forms of NOT8 (Chang et al. 2019) and DCP2 (Loh, Jonas, and Izaurralde 2013) to inhibit deadenylation and decapping.

These experiments provide additional mechanistic details for 5’ shortening and suggest endonucleolytic cleavage as a critical step (Fig. 3 and Sup. Fig. 3). We agree that it would be interesting to study the fate of these shortened transcripts notably regarding translation. However, given the complexity of the expected proteome changes also following global translation arrest under stress (Harding et al., 2003; Pakos-Zebrucka et al., 2016), we think that this work is beyond the scope of this manuscript and will be the subject of future studies.

Minor comments:(1) Some of the affected RNAs can be validated in HeLa and other cell lines.

We thank the reviewer for their suggestion. We have performed RT-qPCR on 3 different mRNAs that present 5’ shortening upon oxidative stress using different primers located along the mRNA. We hypothesized that the closer the primer set is located to the 5’ end, the less abundant the corresponding region would be for arsenite-treated compared to untreated cells. Our results show indeed that the measured level of these mRNAs depends on the location of the primer sets used for the qPCR, the closer to the 5’end it is, the less abundant the mRNA is upon oxidative stress compared to control cells. We present these data as well as a schematic representing the positions of the primers in Sup. Fig. 2d.

(2) The authors should check whether XRN1 also co-localizes in SGs.

We thank the reviewer for their suggestion. We have performed immunofluorescence on U2OS and HeLa upon oxidative stress and did not observe a co-localization of XRN1 with TIA-1, a marker of stress granules (see below). These results are consistent with (Kedersha et al. 2005) that have shown that XRN1 mainly co-localizes to processing bodies and are very weakly detectable in SGs in DU145 cells. We think that this result is beyond the scope of this study and thus decided to only include it for the reviewers.

**Author response image 1. sa4fig1:** Representative immunofluorescence merged image of HeLa (left panel) and U2OS (right panel) cells treated with sodium arsenite and labelled with anti-TIA1 (red), anti-XRN1 (green) antibodies and DAPI (blue). Scale bar 50 µm.

(3) XRN1 should be knocked down with more than one siRNA.

We thank the reviewer for this suggestion. Our results show that our XRN1 KD specifically rescues the length of the most shortened mRNAs (Fig. 3e). This is a highly specific effect that makes us confident it is not mediated by non-specific siRNA binding; thus, we do not consider it necessary to repeat the experiment.

(4) There are typos in the text regarding Figure 6d, e, and f. Also, Supplementary Figure 4a.

We thank the reviewer for identifying these mistakes. We have corrected the typos.

**Reviewer #3 (Recommendations For The Authors):**
The authors should consider testing their hypotheses by arresting the decay pathway using the approaches I mentioned previously. As it stands, some conclusions are somewhat speculative.

We have replied to the reviewer comments in the public review section.

**References:**

Brothers, William R., Farah Ali, Sam Kajjo, and Marc R. Fabian. 2023. “The EDC4-XRN1 Interaction Controls P-Body Dynamics to Link MRNA Decapping with Decay.” *The EMBO Journal*, August, e113933.Chang, Chung-Te, Sowndarya Muthukumar, Ramona Weber, Yevgen Levdansky, Ying Chen, Dipankar Bhandari, Catia Igreja, Lara Wohlbold, Eugene Valkov, and Elisa Izaurralde. 2019. “A Low-Complexity Region in Human XRN1 Directly Recruits Deadenylation and Decapping Factors in 5’-3’ Messenger RNA Decay.” *Nucleic Acids Research* 47 (17): 9282–95.Harding, Heather P., Yuhong Zhang, Huiquing Zeng, Isabel Novoa, Phoebe D. Lu, Marcella Calfon, Navid Sadri, et al. 2003. “An Integrated Stress Response Regulates Amino Acid Metabolism and Resistance to Oxidative Stress.” *Molecular Cell* 11 (3): 619–33.Ibrahim, Fadia, Manolis Maragkakis, Panagiotis Alexiou, and Zissimos Mourelatos. 2018. “Ribothrypsis, a Novel Process of Canonical MRNA Decay, Mediates Ribosome-Phased MRNA Endonucleolysis.” *Nature Structural & Molecular Biology* 25 (4): 302–10.Kedersha, Nancy, Georg Stoecklin, Maranatha Ayodele, Patrick Yacono, Jens Lykke-Andersen, Marvin J. Fritzler, Donalyn Scheuner, Randal J. Kaufman, David E. Golan, and Paul Anderson. 2005. “Stress Granules and Processing Bodies Are Dynamically Linked Sites of MRNP Remodeling.” *The Journal of Cell Biology* 169 (6): 871–84.Krause, Maximilian, Adnan M. Niazi, Kornel Labun, Yamila N. Torres Cleuren, Florian S. Müller, and Eivind Valen. 2019. “Tailfindr: Alignment-Free Poly(A) Length Measurement for Oxford Nanopore RNA and DNA Sequencing.” *RNA* 25 (10): 1229–41.Loh, Belinda, Stefanie Jonas, and Elisa Izaurralde. 2013. “The SMG5-SMG7 Heterodimer Directly Recruits the CCR4-NOT Deadenylase Complex to MRNAs Containing Nonsense Codons via Interaction with POP2.” *Genes & Development* 27 (19): 2125–38.Love, Michael I., Wolfgang Huber, and Simon Anders. 2014. “Moderated Estimation of Fold Change and Dispersion for RNA-Seq Data with DESeq2.” *Genome Biology* 15 (12): 550.Pakos-Zebrucka, Karolina, Izabela Koryga, Katarzyna Mnich, Mila Ljujic, Afshin Samali, and Adrienne M. Gorman. 2016. “The Integrated Stress Response.” *EMBO Reports* 17 (10): 1374–95.Pelechano, Vicent, Wu Wei, and Lars M. Steinmetz. 2015. “Widespread Co-Translational RNA Decay Reveals Ribosome Dynamics.” *Cell* 161 (6): 1400–1412.